SciPost Physics Codebases

# The ITensor Software Library for Tensor Network Calculations

Matthew Fishman[1], Steven R. White[2], E. Miles Stoudenmire[1*]

**1** Center for Computational Quantum Physics, Flatiron Institute, New York, NY 10010, USA

**2** Department of Physics and Astronomy, University of California, Irvine, CA 92697-4575 USA

\* mstoudenmire@flatironinstitute.org

December 18, 2021

## Abstract

**ITensor is a system for programming tensor network calculations with an interface modeled on tensor diagrams, allowing users to focus on the connectivity of a tensor network without manually bookkeeping tensor indices. The ITensor interface rules out common programming errors and enables rapid prototyping of algorithms. After discussing the philosophy behind the ITensor approach, we show examples of each part of the interface including Index objects, the ITensor product operator, tensor factorizations, tensor storage types, algorithms for matrix product state (MPS) and matrix product operator (MPO) tensor networks, quantum number conserving block sparse tensors, and the NDTensors library. We also review publications that have used ITensor for quantum many-body physics and for other areas where tensor networks are increasingly applied. To conclude we discuss promising features and optimizations to be added in the future.**

# 1   Introduction

Tensor networks are a technique for working with tensors which have many indices [1–6]. The naive memory and computing costs of working with a tensor having $N$ indices (an

order-$N$ tensor) scales exponentially with $N$. A *tensor network* is a representation of a large, high-order tensor as the contracted product of many low-order tensors. When all of the tensors in the network are low-order, a tensor network can make it efficient to perform important operations such as summing two high-order tensors or computing their inner product. These operations can remain efficient whether the high-order tensor represented implicitly by the network has hundreds, thousands, or even an infinite number of indices.

Describing tensor networks can be difficult when using traditional notation: one must come up with distinct names for indices and it can be hard to see the connectivity pattern of the network. An elegant alternative is tensor diagram notation [7]. In diagram notation, tensors are shapes and indices are depicted as lines emanating from them. Connecting two index lines means they are contracted or summed over. For example, the following diagram is equivalent to the traditional expression below it:

$$\sum_k T_{ijk} M_{kn} \quad = \quad R_{ijn}$$

Diagram notation is enormously helpful for expressing tensor networks, as it emphasizes key aspects of tensor algorithms while suppressing implementation details such as the ordering of tensor indices. It is just as rigorous as traditional index notation.

ITensor, short for *intelligent tensor*, is a software library inspired by tensor diagram notation. Its goal is enabling users to translate a tensor diagram into code without reintroducing concepts not expressed by tensor diagrams. For example, when summing two ITensors the only requirement is that they have the same set of indices in any order; the ITensor system handles all other details of performing the sum.

Two "philosophical" principles guided the design of ITensor. The first was that in using the library, any implementation details which are not a part of the conceptual algorithms should be kept hidden from the user as much as possible. Not having to think about these details allows one to focus more clearly on the essentials. A key early insight was that this principle could apply to the ordering of indices in an ITensor. In typical tensor software, the user is constantly thinking about the order of the indices. However, tensor diagrams do not have any index ordering, just labels that keep track of the relevant information. Thus ITensors have the ordering of their indices abstracted away as an implementation detail, using "intelligent" indices that retain their identity. The second key principle is that the software should allow one to interact with it at a variety of levels. At a high level, for calculations done in a standard way one can call functions encapsulating a sophisticated algorithm (say, the density matrix renormalization group, DMRG) without understanding much of the implementation details. At an intermediate level, one can gain flexibility by working with moderately sophisticated routines, such as for adding MPS. And finally, to do something more novel, one can work at the lower level of individual ITensors. This multilevel access mandated that ITensor be a library, not a single executable program with complicated input files.

These initial principles led to other interesting design choices over time. One consequence of having intelligent tensor indices as distinct data objects (of type `Index`) is that they can store extra information about themselves. A key use case is indices which have

internal subspaces labeled by conserved quantum numbers (symmetry group representation labels). Storing this information in the `Index` objects ensures that when contracting two quantum number conserving ITensors, both are guaranteed to use the same ordering of the subspaces for storing their data. Another consequence is that dense and sparse ITensors can be of the same type and have essentially the same interface, because the implementation can inspect the indices and internal storage type to determine the actual 'type' of any ITensor. Thus users can write very generic code that works for any type of ITensor.

The design choice to have an ITensor manage its own index ordering is by no means an obvious one. Benefits of ITensor's *intelligent index* system include making addition of ITensors $A$ and $B$ as simple as writing the code $A + B$ for tensors with the same indices, or automating the application of operators to matrix product states (MPS). A system to automate anti-commuting "fermionic" tensor algebras is currently in development which heavily relies on the intelligent index system to keep track of index ordering. Calculations where multiple tensor diagrams have many of the same indices is another example where intelligent indices makes code simpler and less error prone. An example is taking the gradient of a tensor network, which simply involves removing that tensor from the network. But possible drawbacks of the intelligent index approach include occasional extra lines of code to manipulate index properties and some loss of control over low-level details of tensor operations when using very high-level features. However we do offer more advanced features that give complete control over such details.

Most tensor libraries, in contrast, choose to expose the ordering of tensor indices to users who must manage this ordering manually [8]. Such interfaces always give users fine-grained control over details that can affect performance, but can put more of a burden on the user to ensure correctness. While ITensor does not require users to think about the index ordering, it can be manually controlled when needed by calling functions such as `permute` to explicitly permute indices into a specified memory ordering. In tensor contractions, the index ordering of the output tensor can be controlled by supplying it through an in-place contraction function.

Another contrast between ITensor and other tensor libraries relates to how networks consisting of many tensors are handled. In many libraries, higher-level network interfaces are offered by supplying temporary text labels for indices [9–11] or by placing tensors into a graph or network structure and specifying contracted indices through the graph topology [12, 13]. Because ITensors have persistently labeled indices, any collection of ITensors with unique indices already specifies a graph. We are currently taking advantage of this property of ITensors to offer higher-level tensor network abstractions and make use of it in our upcoming automatic differentiation tools.

ITensor was first implemented in C++ and extensively developed and refined through three major releases over ten years. [1] Recently, ITensor has been fully ported to the Julia language and most new features are being developed there. [2] In what follows we show examples in Julia, though we emphasize that the high-level C++ and Julia interfaces are quite similar (see the Appendix for full code examples in each language). Both versions are full implementations of ITensor in each language: the Julia version is not a wrapper around the C++ version.

The goal of this article is to provide a high-level overview of the ITensor system, its design goals, and its main features. Much more information including detailed documentation of the ITensor interface, code examples, and tutorials can be found on the ITensor website: https://itensor.org.

---

[1]ITensor Github Repository (C++): https://github.com/ITensor/ITensor
[2]ITensor Github Repository (Julia): https://github.com/ITensor/ITensors.jl

## 2 Interface Examples

We first introduce ITensor by giving examples as an informal overview. In later sections, we will discuss many more details of the individual elements making up the ITensor system such as "intelligent" tensor indices, tensor factorizations, and block sparse ITensors.

### 2.1 Installing ITensor

Julia features a built-in package manager that makes installing libraries simple. To install the ITensor library, all a user has to do is issue the following commands, starting from their terminal:

```
$ julia
julia> ]
pkg> add ITensors
```

The `julia` command starts an interactive Julia session and typing `]` enters package manager mode. The command `add ITensors` downloads and installs all the dependencies of the ITensors.jl package then finally the ITensor library itself.[3]

### 2.2 Obtaining Help

Once ITensor is installed, the built-in Julia documentation system can be used to query ITensor functions and types. For example

```
julia> using ITensors
julia> ?
help?> Index
```

will give the output

```
search: Index indexin IndexStyle IndexLinear ...

  An Index represents a single tensor index with fixed
  dimension dim. Copies of an Index compare equal unless
  their tags are different.

  ...
```

and additional information describing the `Index` type and its constructors.

### 2.3 Basic ITensor Usage

To begin using the ITensors package in a Julia session or script, input the line

---

[3]The reason the Julia library is called "ITensors" and not ITensor is to keep the module name from conflicting with name of the ITensor type.

```
using ITensors
```

Before creating an ITensor, one first creates its indices. The line of code

```
i = Index(3)
```

creates a tensor Index of dimension 3 and assigns this Index object to the reference `i`. Upon creation, this Index is stamped with an immutable, unique id number which allows copies of the Index to be compared and matched to one another. A portion of this id is shown when printing the Index, with typical example output of the command `@show i` being:

```
i = (dim=3|id=804)
```

After making a few Index objects `i,j,k,l` one can define ITensors:

```
A = ITensor(i)
B = ITensor(j,i)
C = ITensor(l,j,k)
```

Because matching Index pairs can automatically recognize each other through their id numbers, tensor contraction can be carried out as:

```
D = A * B * C
```

The `*` operator finds all matching indices between two ITensors and sums over or contracts these indices. The `i` Index is summed in the first contraction above and `j` in the second, leaving `D` with indices `l` and `k`. The ITensor product operator "`*`" can also be used for outer products and scalar products, and is discussed in more detail in Section 4.

## 2.4 Setting ITensor Elements

Setting an element of an ITensor `A = ITensor(i,j,k)` is done by

```
A[i=>2,j=>3,k=>1] = 0.837
```

which assigns the value 0.837 to the element of `A` for which index `i` has the value 2, `j` has value 3, and `k` has value 1. (Note that in Julia, the built-in notation `x=>y` makes a `Pair(x,y)` object.) ITensor indices are 1-indexed, similar to Julia arrays.

Because the Index objects are provided along with their values, they can be passed in any order. Thus the following lines of code

```
A[i=>2,j=>3,k=>1] = 0.837
A[k=>1,i=>2,j=>3] = 0.837
```

have exactly the same effect on the ITensor A.

To create an ITensor with normally-distributed random elements instead of specific values, one can use the constructor

```
T = randomITensor(i,j,k)
```

to make a real-valued random tensor or

```
T = randomITensor(ComplexF64,i,j,k)
```

to construct a complex-valued random ITensor.

## 2.5 Matrix Example

To illustrate the usefulness of the ITensor approach involving Index objects and the `*` operator, consider a pair of order-2 tensors (matrices)

```
A = ITensor(i,j)
B = ITensor(k,j)
```

In a typical matrix or tensor library, to contract A with B and sum over their shared index j, one would need to write code similar to

```
C = A * transpose(B)
```

Note that the above line is *not* ITensor code!

Within ITensor, all one needs to do is to write

```
C = A * B
```

and the `*` operator handles the transposition of B automatically. If B is redefined with the ordering of its indices reversed, the operation A `*` B continues to give the correct result. This type of behavior makes ITensor applications robust to changes in the code that may modify the ordering of tensor indices or the layout of tensors in memory.

## 2.6 Summing ITensors

ITensors can be added and subtracted as long as they have the same set of sindices. Even if the indices are in a different order, addition always works straightforwardly because the ITensor system is able to internally deduce the data permutation required:

```
A = randomITensor(i,j,k)
B = randomITensor(k,i,j)
C = A + B
```

ITensors may also be subtracted and multiplied by scalars, including complex scalars, for example:

```
D = 4*A - B/2
F = A + 3.0im * B
```

## 2.7  Priming Indices

Sometimes it is not desirable to contract all of the indices shared between two tensors. Consider two ITensors

```
A = ITensor(i,j)
B = ITensor(i,j)
```

and say we want to contract only over the index `j` leaving the `i` indices uncontracted.

A convenient way to achieve this while still using the `*` operator is to *prime* one of the `i` indices

```
Ap = prime(A,i)
```

The ITensor `Ap` has the same elements as `A` but has indices `(i',j)`. When contracting `Ap` with `B`, now only the `j` indices will match or compare equal, so it will be the only Index contracted

```
C = Ap * B
hasind(C,i) == true
hasind(C,i') == true
```

Diagrammatically we can notate the above contraction as:

## 2.8 Compiling ITensor

Although the experience of using Julia is similar to using an interpreted language, it is actually a *just-in-time compiled* language.

The initial compilation time when Julia first encounters new functions or types can be large in a new Julia session, though there is ongoing work to provide ahead-of-time compilation tools for Julia. To reduce just-in-time compilation overhead, we offer a convenient way for users to compile most of the ITensors.jl code ahead of time with the following commands within an interactive Julia session:

```julia
julia> using ITensors
julia> ITensors.compile()
```

The compilation process can take many minutes, but only has to be performed once each time the ITensors.jl library is upgraded to a new version. After the command is run, it will suggest command-line arguments that can be passed to the `julia` language program that will load a precompiled ITensors.jl system image when running Julia. Running ITensor code this way typically reduces startup times to only a few seconds.

## 2.9 Online Code Examples

For more extensive and frequently updated examples of ITensor code, including full applications, we include an set of examples as part of our source code distribution at the following link: ITensor Code Examples.

# 3 Index Objects

A core concept of the ITensor system is that tensor indices carry information beyond just their dimension. Mathematically, this corresponds to the notion that an index labels the basis of a vector space, and that two vector spaces may be different from each other despite having the same dimension.

The notion that a tensor index corresponds to a specific vector space is encoded in the unique *id number* assigned to an Index object when it is constructed:

```julia
i = Index(4)
@show i  # prints: i = (dim=4|id=577)
```

Printing an Index as in the code above shows a portion of the (64 bit) id number.[4]

Because a new id is assigned each time an Index is constructed, other separately constructed Index objects will not compare equal to `i` even if they have the same dimension

```julia
j = Index(4)
j != i  # true
```

---

[4]As a technical note, the Index id numbers are generated randomly, but collisions are highly improbable because of the 64-bit length of the ids. Random id generation has many advantages over sequential, including using ITensor for parallel algorithms and reading ITensors from files and mixing these ITensors with newly generated ones.

In other words, comparison operations (`==`,`!=`) require two Index objects to have the same id for them to compare equal.

To enrich the Index system one may also add tags to indices

```
s = Index(3,"s,Site")
```

The Index `s` above has a dimension 3, as well as two tags `"s"` and `"Site"`. For efficiency reasons, tags can have a maximum of eight characters and indices can have a maximum of four tags. These maximum values are currently hard-coded into the library and may be increased in the future as use cases arise that require longer tags or more tags.

Tags can serve multiple purposes: helping to identify Index objects when printing them; collecting subsets of indices sharing a common tag or tags; and preventing certain Index pairs from contracting with each other. This last use of tags extends the rule for Index comparisons: for Index objects to compare equal they must have the same tags as well as the same id number.

As discussed in the previous section, one other way to prevent Index objects from comparing equal is to change their *prime level*. Every Index carries an integer prime level which defaults to zero.

```
i = Index(2,"i")
@show plev(i) # plev(i) = 0
```

A copy of Index `i` but with a prime level of 1 can be created by calling

```
ip = prime(i)
@show plev(ip) # plev(ip) = 1
```

or for convenience by writing

```
ip = i'
```

Two copies of the same Index which have different prime levels do not compare equal

```
i == i'  # false
i == i'' # false
```

Because both primes and tags can be used to prevent Index objects from comparing equal to each other and being contracted by the `*` operator, some experience is needed to choose the best approach. Primes are useful when indices are only distinguished temporarily; it is easy afterward to call `noprime(T)` on an ITensor to reset the prime levels of all of its indices. On the other hand, tags should be used when there is some application-specific

understanding of why certain indices are distinguished. For example in the case of a tensor network with a square lattice structure, where all indices linking the tensors together may describe the same vector space, we might use the tags `"x=-1"`, `"x=0"`, `"x=1"`, ... to label a unique horizontal position in the lattice and the tags `"y=-1"`, `"y=0"`, `"y=1"`, ... to specify a unique vertical position. This is particularly useful in applications involving translational invariance, where many copies of the same Index can appear in different contexts and it can become cumbersome to distinguish them by prime levels alone.

# 4 The ITensor Product Operator (∗)

Just as tensor diagrams unify many concepts, the ITensor product operator ⋆ likewise unifies many operations into a single operation:

- The ⋆ product of ITensors with no indices in common computes an *outer product.*

- The ⋆ product of ITensors with all the same indices computes an *inner product,* resulting in a scalar ITensor.

- Otherwise, for a pair of ITensors having just some indices in common, the ⋆ operator computes a *tensor contraction.*

A simple example of an outer product is the product of two vectors which do not share a common index:

```
v = ITensor(i)
w = ITensor(j)
x = v * w
```

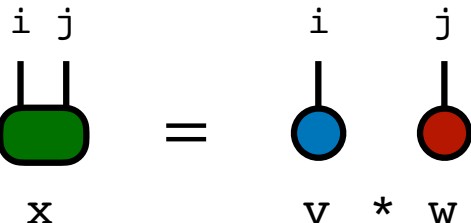

Using the ⋆ operator to compute an inner product results in a scalar ITensor with no indices as in the following example (note that the indices do not need to be in the same order for the result to be correct):

```
A = ITensor(i,j,k)
B = ITensor(k,i,j)
C = A * B
```

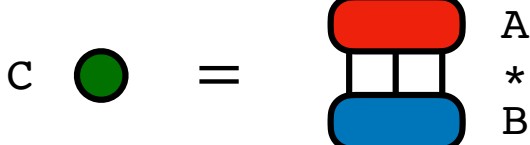

The `scalar` function can be called to convert a scalar ITensor into a real or complex number

```
x = scalar(C)
```

or alternatively one can call `x = C[]`.

Finally, to illustrate the case of a tensor contraction where only some of the indices are summed, we can use the following example which was also shown at the beginning of this article:

```
T = ITensor(i,j,k)
M = ITensor(k,n)
R = T * M
```

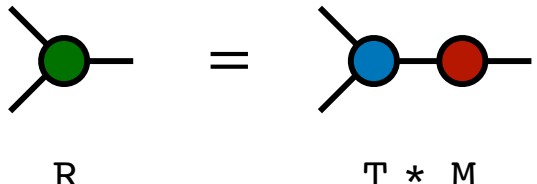

In the diagram above, we have omitted the names of the indices to emphasize the typical user experience: all that a user needs to know to get a correct result in the above example is that `T` and `M` share one Index. Keeping track of the ordering of the uncontracted indices, which become the indices of `R`, is not necessary.

Besides contracting regular tensors, the `*` operator can also be used in conjunction with specially constructed tensors to manipulate tensor indices. One example of such a special tensor type is a *delta tensor*, also known as a copy tensor, which has all diagonal elements equal to one and other elements equal to zero, and is often shown as a solid black circle in tensor diagrams. In the ITensor library, a delta tensor uses special diagonal-sparse storage internally, not only to save memory but also to ensure that the contraction of delta tensors with other tensors is performed using specially optimized routines.

A delta tensor can be used to replace an Index with another Index of the same dimension:

```
A = ITensor(k,j)
A = A * delta(k,i)
@show hasind(A,i) # true
```

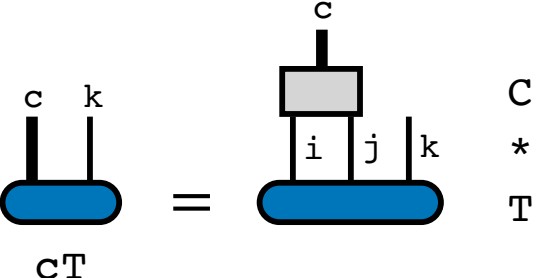

or to duplicate (or split) an index as follows:

```
B = ITensor(k)
B = B * delta(k,i,j)
```

Note that in Julia, one can use the unicode character $\delta$ to write the code above as `B = B * δ(k,i,j)`.

Another example of a special tensor type is a *combiner* ITensor. When contracted with another ITensor, a combiner merges multiple indices into a single Index.

```
T = ITensor(i,j,k)
C = combiner(i,j)
cT = C * T
```

The Index `c` shown in the diagram above can be retrieved by calling `combinedind(C)` on the combiner ITensor. Alternatively one can call `commonind(C,cT)` to retrieve this Index, since it is the one that the combiner and `cT` will necessarily have in common.

Taking the product with the conjugate of the combiner reverses this operation.

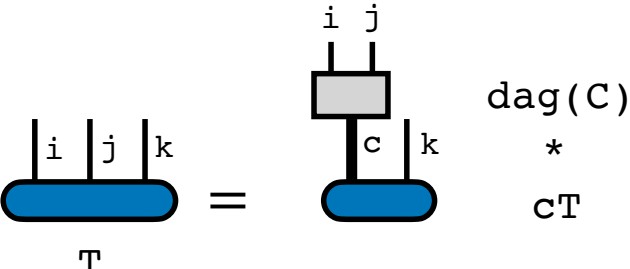

Like delta tensors, combiners also use a special storage type with a negligible memory footprint and optimized contraction algorithms for combining and uncombining indices.

The action of a combiner on a tensor is conceptually identical to the concept of permuting and reshaping a multi-dimensional array, at least for the case of dense ITensors. For quantum number conserving or symmetric ITensors, combiners can perform additional steps like grouping multiple copies of a quantum number together in the combined Index, or managing anticommutation properties in the case of the upcoming ITensor fermion system.

# 5    Tensor Decompositions

Many commonly used tensor network decompositions are built from matrix decompositions such as the QR and singular value decompositions (SVD) known from linear algebra. Despite being defined in terms of matrices, these factorizations can be straightforwardly defined for tensors too. All that is needed is a mapping from a tensor to a matrix, defined by specifying a certain group of indices as row indices and the rest as column indices, then treating each group as a single larger index when computing the decomposition. ITensor automates the tedious and error-prone process of converting tensors to matrices and back, providing a tensor-level interface for various decompositions.

Consider an ITensor `T` with indices `i,j,k`. We can compute a QR decomposition of `T` by just specifying that `i,k` are the row indices as follows:

```
T = randomITensor(i,j,k)
Q,R = qr(T,(i,k))
```

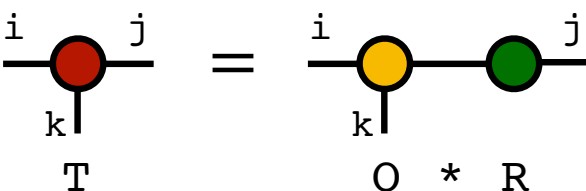

A new Index is generated by the `qr` function which links the `Q` tensor to the `R` tensor as shown above. This makes it straightforward to recover the tensor `T` just by using the `*` operator:

```
Q*R ≈ T  # true
```

(In Julia, the ≈ operator is overloaded to compute the relative difference between the two sides of an equation, and return true if it is below a prescribed threshold.) Note that when computing the product `Q*R` one does not need to know any details of the new Index introduced by the factorization, such as whether it is the first or second index of R, or its dimension. However, in situations where one wants to retrieve this Index, a convenient way to do it is as follows:

```
q = commonind(Q,R)
```

where the `commonind` function returns the first Index found that is shared by the two ITensors.

The SVD plays a key role in tensor network calculations, and is implemented as

```
W = randomITensor(i,j,m,k)
U,S,V = svd(W,(j,i))
U*S*V ≈ W  # true
```

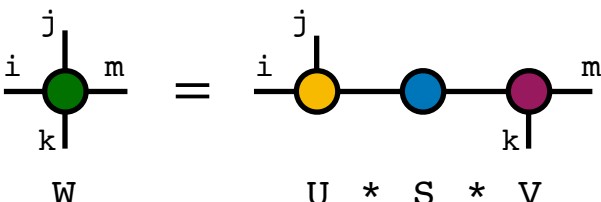

In the example above, `j,i` were specified as the row indices, leaving `m,k` as the column indices.

An important feature of certain decompositions such as the SVD is that they allow controlled truncation of the tensors resulting from the factorization. By default, ITensor decompositions do not truncate, though they do always compute the "thin" version of a decomposition when available. A truncated decomposition can be computed by specifying truncation keyword arguments. In the following example

```
U,S,V = svd(W,(j,i);cutoff=1E-8,maxdim=10)
```

the truncation will be determined by summing the squares of the singular values from smallest to largest until the truncation error reaches $10^{-8}$ while also ensuring that the maximum number of singular values kept is less than or equal to 10.

# 6  Tensor Storage Layer

A powerful feature of ITensor is that ITensors can have a wide variety of storage formats while offering the same user interface. Users can mix sparse and dense tensors together in calculations and manipulate any kind of tensor using identical high-level code.

In most cases users do not set the storage type manually; instead special storage types occur automatically when using other features: after computing the singular value decomposition of an ITensor, the singular values are returned as an ITensor with diagonal-sparse storage; constructing an ITensor from indices with quantum number subspaces makes the storage automatically block sparse.

Importantly, because the storage types used by an ITensor are distinct types, each one can use the most optimal memory layout possible, and performance-critical algorithms such as tensor contraction and factorization can be specialized for each storage type or combination of storage types. For this purpose, we take full advantage of Julia's multiple dispatch mechanism to organize specialized algorithms into separate code pathways to keep the library code simple. These optimizations are hidden from the user, who can just contract ITensors together using the `*` operation and automatically get the best possible performance available.

Some of the most common storage types available in ITensor are:

- *Dense storage*: this is the default storage type when constructing an ITensor from regular Index objects and setting elements. The Dense storage type is parameterized over its element type, so that `Dense{Float64}` (real-valued dense storage) and `Dense{ComplexF64}` (complex-valued dense storage) are actually different storage types. The type used to hold the data for Dense storage can also be changed through a second, optional type parameter, to types such as `Vector{Float64}` or `SubArray{Float64}`.

- *Diagonal storage*: diagonal-sparse tensors occur naturally in algorithms such as the singular value decomposition and eigenvalue decomposition. In such settings, all of the diagonal elements can be different and so an array of the diagonal elements is stored. A special case of diagonal storage is uniform diagonal storage, where all of the elements of the diagonal are constrained to be the same. For this special storage only the value of the repeated, identical diagonal element is stored and specially-optimized contraction algorithms are invoked. If the uniform diagonal value is equal to `1.0` then such a diagonal tensor can be used to replace one Index with another under the contraction or `*` operation, or as a "copy" or "delta" ($\delta$) tensor as used in certain tensor network algorithms.

- *Combiner storage*: This storage type uses essentially no memory and stores no tensor components. Rather, it stands for a tensor which conceptually merges two or more indices into one larger index. A combiner tensor `C` can be created as `C = combiner(i,j,k)` where `i,j,k` are the indices one wants to combine together. Contracting the combiner ITensor with an ITensor having these indices results in a new ITensor where the indices are merged into the Index `cind = combinedind(C)`. The new combined Index is created automatically by the combiner.

- *Block sparse storage*: Block sparse storage is automatically used when an ITensor is created from Index objects with quantum number subspaces. This is an important case for quantum physics calculations, where the sparsity enforces symmetries or conservation laws and allows calculations to be performed more efficiently. The block sparse and quantum number system is discussed in more detail in Section 8.

An important consideration for block sparse storage is that the overhead of managing the layout of blocks and movement of blocks within algorithms must be kept very low in order to benefit from the efficiency of the tensor sparsity. Currently, the ITensor block sparse storage holds all of the non-zero tensor elements in a single, contiguous array and keeps a dictionary mapping block indices such as `(2,1,7)` to offsets in the array.

- *GPU storage*: GPU (graphics processing unit) storage is an experimental feature supported by the ITensorGPU.jl package. An ITensor with GPU storage stores its elements in GPU memory, and calls specialized routines for operations including tensor contraction and tensor factorizations. Taking advantage of the parallel processing capabilities of GPUs can give speedups ranging from two to a hundred times the speed of CPU calculations. Because different storage types are handled automatically behind the same ITensor interface, GPU ITensors can take advantage of the same set of high-level algorithms available in the ITensor library written originally for regular tensors stored in host memory.

- *Empty Storage* : ITensors support a special storage type `EmptyStorage` which is used to represent an ITensor which is numerically zero but without incurring the cost of allocating any memory. Calling a constructor such as `ITensor(i,j,k)` results in an ITensor with empty storage.

  Another feature of the empty storage type is that it can be used as a convenient workaround for specifying a complicated set of tensor indices in advance. A key example is when summing a set of tensors which are known to have the same indices as each other, but where the user does not want or need to explicitly work with these indices. In such cases, a default-initialized ITensor (which will have empty storage) can be used as a "universal zero" tensor which can be summed with any other tensor, for example:

```
i = Index(2)
V = [randomITensor(i), randomITensor(i)]
T = ITensor()
for A in V
  T += A
end
```

The flexibility of the ITensor storage system will let us explore other interesting possibilities in the future. Some planned extensions include `IdentityStorage` storage which represents an identity map from one collection of indices to another, `UnitaryStorage` storage representing a unitary map, and storage types which handle common operations such as conjugation in a lazy or delayed manner.

An important direction we plan to pursue is further sparsity patterns, including fully general sparsity. Technically, general sparsity is already handled by the ITensor block sparse system in the limit of all block sizes set to 1, and we have already observed speedups from representing sparse tensors in this limit. However, we plan to expose generally sparse tensors more explicitly and possibly handle them in a more optimized way.

Lower-precision floating-point data is already supported by our storage layer, and can significantly speed up calculations such as when using GPU hardware. Also we have experimental support for more exotic numerical types such as *tropical* numbers, thanks to

contributions by Jin-Guo Liu [14]. More systematic handling of numerical types such as integer, boolean, or nonnegative tensor elements is a planned future direction.

Given the usefulness of the flexible storage type system in ITensor, we plan to formalize and carefully document the steps for users to make their own custom storage types. Because of the dynamic nature of the Julia language, such types can be fully defined outside of the ITensor library itself yet be treated as first-class storage types for ITensors.

# 7 High Level Features: MPS and MPO Algorithms

To make ITensor a productive system for rapidly prototyping tensor network algorithms, it provides the most common and well-developed tensor network formats and algorithms. The two most well developed formats are the matrix product state (MPS) tensor network [4, 15, 16], also known as the tensor train [6], and the matrix product operator (MPO) tensor network [17, 18].

Algorithms included with the core ITensor library include summation of MPS and MPO; truncation of MPS and of MPO; optimization of MPS through the DMRG algorithm; and multiplication of an MPS by an MPO. These algorithms offer a high degree of customizability: the multiplication of an MPS by an MPO can be performed using at least three different algorithms (selected by a keyword argument), with each algorithm offering tradeoffs in terms of scaling, performance, and controllability. The DMRG code offers different modes, including finding the ground state (dominant eigenvector) of an implied sum of multiple MPOs or finding excited states (sub-dominant eigenvectors).

Throughout this section, code examples will use strings denoting local operators such as `"Sz"`, `"S+"`, or `"S-"` or strings denoting states of the local Hilbert space such as `"Up"` and `"Dn"`. The way ITensor is able to know the appropriate definition of these operators and states is through a flexible and extensible system of mapping operator and state names to tensors and tensor elements.

## 7.1 OpSum and AutoMPO

A very useful and popular feature of ITensor is the OpSum/AutoMPO system. An OpSum is a type that lets users input sums of products of local linear operators in a domain-specific language and AutoMPO is the backend system for "compiling" these sums to MPO tensor networks. Constructing sums of local operators is particularly important for physics applications, where one studies Hamiltonian operators. A typical example being the Heisenberg Hamiltonian:

$$H = \sum_{i=1}^{N-1} \vec{S}_i \cdot \vec{S}_{i+1} = \sum_{i=1}^{N-1} S_i^z S_{i+1}^z + \frac{1}{2} S_i^+ S_{i+1}^- + \frac{1}{2} S_i^- S_{i+1}^+ \quad . \tag{1}$$

This particular Hamiltonian can be exactly written as an MPO of bond dimension 5, [18] but the construction is technical and tedious to program by hand. The AutoMPO system automates the construction of this Hamiltonian MPO from the OpSum object:

```
function heisenberg_mpo(N)
  # Make N S=1/2 spin indices
  sites = siteinds("S=1/2",N)

  # Input the operator terms
```

```
  os = OpSum()
  for i=1:N-1
   os +=     "Sz",i,"Sz",i+1
   os += 1/2,"S+",i,"S-",i+1
   os += 1/2,"S-",i,"S+",i+1
  end

  # Convert these terms to an MPO
  H = MPO(os,sites)

  return H
end

H = heisenberg_mpo(100)
```

Comparing the lines of code in the for loop above to the Hamiltonian definition in Eq. (1) one can observe a close similarity.

The AutoMPO system is powerful. Following a major enhancement of the backend code by Anna Keselman based on Ref. [19], AutoMPO can accept terms with more than two local operators and local operators separated by arbitrary distances, and uses an SVD-based compression algorithm to obtain a nearly-optimal MPO bond dimension.

We are working on or envision many useful extensions to this system:

- Improved compression techniques based on better-scaling algorithms generalized from techniques used for non-local Hamiltonians arising in quantum chemistry [20].

- Compiling exponentials of OpSums into quantum circuits using Trotter-Suzuki decompositions.

- Extensions to infinite, translation-invariant systems, including truncation methods developed for infinite MPOs like the ones introduced in Ref. [21].

- Generalizations to other tensor network topologies, such as tree tensor networks (TTNs) and projected entangled pair operators (PEPOs).

- Converting OpSums corresponding to interacting fermionic Hamiltonians to free fermion approximations using mean field approximations like Hartree-Fock, which could then be used by free fermion formulations of tensor networks [22] [5].

## 7.2 DMRG Algorithm

One of the most heavily used high-level algorithms included with ITensor is the density matrix renormalization group (DMRG) [3,23]. The DMRG algorithm computes low-energy states of quantum systems, or in mathematical terms, dominant eigenvectors of very large Hermitian linear operators.

The main inputs to a DMRG calculation is a Hamiltonian $\hat{H}$ and an initial guess $\Psi_0^{(i)}$ for its ground state $\Psi_0$. The ITensor DMRG implementation works generically for any Hamiltonian which can be represented as an MPO tensor network, so that the same code can be applied not only to one-dimensional systems, but also quasi-two-dimensional systems and systems with long-range interactions. By taking advantage of the OpSum system discussed above, users can rapidly set up DMRG calculations of complicated Hamiltonians.

---

[5] ITensorGaussianMPS.jl is a package for constructing tensor networks of free fermion states.

Given a Hamiltonian MPO constructed as in Section 7.1 above, one can prepare an initial product state, a schedule of sweeps (DMRG algorithm iterations) and accuracy parameters, then run the DMRG algorithm:

```
# Prepare initial state MPS
state = [isodd(n) ? "Up" : "Dn" for n=1:N]
psi0_i = MPS(sites,state)

# Do 10 sweeps of DMRG, gradually
# increasing the maximum MPS
# bond dimension
sweeps = Sweeps(10)
setmaxdim!(sweeps,10,20,100,200,400,800)
setcutoff!(sweeps,1E-8)

# Run the DMRG algorithm
energy,psi0 = dmrg(H,psi0_i,sweeps)
```

For Hamiltonians defined as the sum of different sets of terms $\hat{H} = \hat{H}_1 + \hat{H}_2 + \hat{H}_3$ one can run a DMRG calculation as:

```
energy,psi0 = dmrg([H1,H2,H3],psi0_i,sweeps)
```

where `H1,H2,H3` are separate MPOs. Instead of summing these MPOs explicitly, which can be costly and inaccurate, the algorithm loops over them internally as if they were summed. This technique can be helpful in applications such as quantum chemistry where Hamiltonians can become large and complex, yet have a nearly block diagonal MPO form if represented as a single MPO. Expressing a Hamiltonian as a sum of MPOs also has the advantage that parts of the DMRG algorithm, like forming the environment tensors and diagonalizing the local effective Hamiltonian, become trivially parallelizable [24]. In initial tests we found that this parallelization is very effective and can be used in conjunction with block sparse parallelism, which we plan to make available as a feature in future versions of ITensor.

To compute an excited state of a Hamiltonian (sub-dominant eigenvector) with ITensor DMRG having first computed both the ground state MPS `psi0`, and first excited state `psi1`, say, one provides `[psi0,psi1]` as an extra argument to DMRG, meaning that the next state computed should be constrained to be orthogonal to these previous ones:

```
energy,psi2 = dmrg(H,[psi0,psi1],psi2_i,sweeps)
```

In the implementation of this particular DMRG routine, projectors onto the previous states `psi0` and `psi1` are effectively added to the Hamiltonian times an "energy penalty", pushing up the energy of these states in the eigenvalue spectrum so they are no longer part of the low-energy subspace [25]. Other techniques for computing excited states are planned in the future, such as the quasiparticle MPS ansatz [26, 27].

## 7.3 MPS and MPO Operations

Far from being black-box software for performing calculations with MPS, ITensor provides many elementary building blocks for creating custom algorithms involving MPS, MPOs, and other tensor networks built from these components such as projected entangled pair states (PEPS).

The most elementary interface to MPS and MPO tensor networks involves retrieving and updating individual factor tensors making up the network. An MPS is a factorization of a tensor psi of the following form

$$\psi^{s_1 s_2 \cdots s_N} = \sum_{\{\alpha\}} A^{s_1}_{\alpha_1} A^{s_2}_{\alpha_1 \alpha_2} A^{s_3}_{\alpha_2 \alpha_3} \cdots A^{s_N}_{\alpha_N} \, , \tag{2}$$

where we have omitted an explicit site-label $j$ on each $A$ tensor for compactness. The factor tensor $A^{s_j}_{\alpha_{j-1} \alpha_j}$ on site $j$ can be obtained as

```
A = psi[j]
```

and updated as

```
psi[j] = new_A
```

To analyze the properties of an MPS, one is often interested in expected values of local operators. To compute the expected value of an operator at every site and return an array of the results, one can use the function `expect`. For example, calling

```
avgSz = expect(psi,"Sz")
```

on an MPS `psi` will compute $\langle\psi|\hat{S}^z_j|\psi\rangle$ for every site $j$ and return an array of the results, where here we use the example of the spin $\hat{S}^z$ operator as our local operator.

Another common quantity of interest is the two-point correlation function of a pair of local operators acting at distant sites $i$ and $j$. Using the example of a spin system again, let us say we are interested in the correlation matrix given by $C_{ij} = \langle\psi|\hat{S}^+_i \hat{S}^-_j|\psi\rangle$. This correlation matrix can be efficiently computed as:

```
C = correlation_matrix(psi,"S+","S-")
```

The `correlation_matrix` function accepts optional keyword arguments such as a smaller range of sites over which to compute the correlation matrix, versus the whole system. It also automatically ensures correct results for fermionic operators such as `"Cdag"` and `"C"` (spinless fermion $\hat{c}$ and $\hat{c}^\dagger$ operators).

An important technical step involving an MPS is bringing it into an orthogonal form, where all of the factor tensors to the left or right of the *center tensor* at a site $j$ are equivalent to partial isometries (i.e. either their rows or their columns are orthogonal). To bring an MPS into orthogonal form efficiently in ITensor, one calls:

```
orthogonalize!(psi,j)
```

where we follow the convention adopted in Julia programming that functions whose name end with ! may modify their first argument. An interesting feature of ITensor MPS objects is that they store information about which tensors are known to be orthogonal, so that calling orthogonalize!(psi,j) repeatedly for the same value of j does no extra work, and shifting the orthogonality center of an already partially orthogonalized MPS can be done with the minimum amount of computation.

Another fundamental operation is truncating an MPS: computing another MPS of a smaller bond dimension which is as close to the original MPS as possible. For MPS such a truncation can be done optimally through various deterministic algorithms. Truncating an MPS psi in ITensor can be done by calling:

```
truncate!(psi;maxdim=500,cutoff=1E-8)
```

where for the sake of example we have shown specific values of the two most commonly used truncation parameters. The maxdim parameter sets an upper limit on the bond dimension of the MPS after the truncation, whereas the cutoff parameter allows the new bond dimension to be determined adaptively as long as the resulting truncation error remains below the value provided. Using a cutoff can allow the bond dimension to fall below the maxdim when possible while still ensuring an accurate approximation of the original MPS.

ITensor supports arithmetic involving MPS and MPOs to be performed using the add function. Performing exact sums can lead to quickly growing costs, so that one normally truncates the result by providing a truncation-error cutoff. For example, to add two MPS psi and phi one can call:

```
eta = add(psi,phi;cutoff=1E-10)
```

and similarly for adding two MPOs. Currently this method uses a particular backend algorithm known as the "density matrix" algorithm [28] but other backends will be available in the future to select through an optional keyword argument.

Algorithms such as time-evolving quantum states or contracting two-dimensional "PEPS" tensor networks can be formulated in terms of products of an MPO with MPS or with another MPO. To approximately multiply an MPS psi by an MPO W, one can call the function

```
Wpsi = contract(W,psi;maxdim=50)
```

with example parameters controlling the truncation shown. The product of two MPOs R and W can also be computed:

```
RW = contract(R,W;cutoff=1E-9)
```

Importantly, these functions provide multiple backend algorithm implementations with various tradeoffs in terms of the cost, accuracy, and control offered. For example, to select the accurate yet expensive "naive" algorithm for multiplying an MPS by an MPO one may call

```
Wpsi = contract(W,psi;method="naive")
```

# 8 Quantum Number Block Sparse ITensors

An important technique used in state-of-the-art physics calculations is enforcing constraints on tensors arising from conserved quantities. These are quantities such as total particle number or total spin along an axis which are conserved due to symmetries of the Hamiltonian operator. The value of each conserved quantity is known as a *quantum number.*

Quantum number conservation can be important since physical systems commonly respect symmetries such as rotational symmetry or particle number conservation symmetry, making it necessary for simulations to conserve these to be comparable to experimental results. Just as importantly, conserving quantum numbers allows calculations to run much faster and use less memory because of a *block sparse* structure that is naturally imposed on the tensors in a tensor network [29]. A detailed discussion of structures imposed by symmetries on tensors and tensor networks is given in Refs. [29–31].

The power of the ITensor approach to conserving quantum numbers is that quantum number conserving ITensors offer nearly the same interface as regular, dense ITensors. Algorithms can be written generically for dense ITensors and automatically work for the symmetric case too, as long as tensors are correctly conjugated using the `dag` function, which would be necessary to use to obtain correct results with complex-valued tensors anyway.

The design of the ITensor quantum number (QN) system is that QN information is stored in Index objects in a fixed order. This information is queried when an ITensor is constructed to determine whether the storage should be block sparse, as well as the layout of the blocks, and which blocks are allocated. When such ITensors are summed, contracted, or factorized, optimized routines are used and the QN information is propagated to the indices of the resulting ITensor.

Currently ITensor only supports quantum numbers arising from symmetries under Abelian groups such as $U(1)$ or $\mathbb{Z}_n$, which are ubiquitous in physics. We are also in the planning stages of support for non-Abelian symmetries such as $SU(2)$ in a future version of ITensor, but the remainder of this section will discuss only the Abelian case.

As an illustrative example of ITensor's QN system, say we have defined two indices with information about their QN subspaces:

```
i = Index(QN(0)=>2,QN(1)=>3;tags="i")
j = Index(QN(1)=>2,QN(2)=>1;tags="j")
```

The Index `i` has a total dimension of 5 because it has two subspaces, one carrying a quantum number `QN(0)` and of dimension of 2; the other carrying a quantum number `QN(1)` and of dimension 3. Similarly `j` has a total dimension of 3, coming from its two

subspaces.

Using these indices, we can define an ITensor `T` in the usual way as

```
T = ITensor(i,j)
```

where initially this ITensor will have Empty storage (see Sec. 6), and thus an as-yet unspecified pattern of non-zero blocks. Then, we set an element of `T` as

```
T[i=>3,j=>1] = 31.0
```

Note that this element corresponds to the `QN(1)` subspace of `i` and the `QN(1)` subspace of `j`, for a combined "QN flux" of `flux(T) == QN(2)`. (Mathematically the flux corresponds to the overall irreducible representation under which the tensor transforms. More intuitively, it describes whether a tensor is a source or sink of quantum numbers and by how much.) When setting any further elements, only those elements of `T` consistent with a flux `QN(2)` will be allowed to be non-zero. This constraint imposes a block-sparse structure on `T`, since most values of the indices combine to form fluxes other than `QN(2)` and thus remain zero. Only allowed blocks consistent with the total flux are stored in memory. Block-sparse computations can then be much more efficient than with dense tensors because fewer non-zero elements have to be handled and the presence of disjoint blocks allows major parts of calculations to be performed in parallel.

For the rest of this section, we discuss in more detail the different types composing the QN block sparse ITensor system.

## 8.1 QN Objects

Block-sparse ITensors arise from vector spaces which are a direct sum of smaller subspaces. In physics calculations, these subspaces are associated with different *quantum numbers*. In ITensor, sets of quantum numbers are stored in QN objects as a collection of name-value pairs, where the value is always an integer. Different values may be combined according to the usual rules of integer addition and subtraction, possibly modulo some other integer $N$. (For the case of quantum numbers arising from non-Abelian symmetries, these rules must be generalized.)

QN objects carrying a single quantum number, such as total z-component spin `"Sz"`, may be constructed as:

```
q0 = QN("Sz",0)
q1 = QN("Sz",1)
```

QNs may be added, subtracted, and compared:

```
q0 + q1 == QN("Sz",1) # true
q1 + q1 == QN("Sz",2) # true
```

QN objects can also carry multiple quantum numbers as follows:

```
a = QN(("N",0),("Sz",0))
b = QN(("N",1),("Sz",-1))
```

Because the quantum numbers are named, they can be provided to the QN constructor in any order and are sorted internally. For convenience when there is only one quantum number, its name can be omitted; this is equivalent to choosing the name to be the empty string.

Some quantum numbers of physical systems obey a $\mathbb{Z}_N$ addition rule. A key example is fermion parity, which is only conserved modulo two in systems such as superconductors. A $\mathbb{Z}_N$ addition rule for a quantum number can be specified by providing $N$ as the third entry of the tuple defining that quantum number:

```
p0 = QN("P",0,2)
p1 = QN("P",1,2)
p1 + p1 == QN("P",0,2)
```

The `2` following the quantum number values above specifies that the `"P"` quantum number obeys $\mathbb{Z}_2$ addition.

The reason quantum numbers have names and are not just distinguished positionally is that having names allows QNs containing different quantum numbers to be combined automatically and correctly. This becomes important when different local physical spaces (such as spin versus particle degrees of freedom) are defined separately, then combined or mixed later. Key examples of physical models combining two otherwise separate types of physical spaces are the Hubbard-Holstein model, where electron sites are intermixed with boson sites, or the Kondo model mixing electron sites with spin sites.

## 8.2 QN Index

As discussed above, the block-sparse structure of quantum number conserving tensors arises from the direct-sum structure of the vector spaces over which they are defined. To specify additional information about direct-sum subspaces, an `Index` object can be constructed from QN-integer pairs, as follows:

```
i = Index(QN("N",0)=>1,
          QN("N",1)=>3,
          QN("N",2)=>2; tags="i")
```

where we note that `(a=>b) == Pair(a,b)` is built-in Julia notation for constructing a pair of values `a` and `b`.

In the example above, the Index `i` has three subspaces, of dimensions 1, 3, and 2 respectively. Therefore the total dimension of `i` is six, or `dim(i) == 6`. The subspaces are associated with the quantum numbers `QN("N",0)`, `QN("N",1)`, and `QN("N",2)` respectively.

A crucial aspect of QN Index objects not yet discussed is that they have an `Arrow` direction, which can be `Out` or `In`, with `Out` being the default. Mathematically, the direction of an Index says whether it is covariant (`In`) or contravariant (`Out`) and expresses

how the Index transforms under the symmetry group action. A physicist might view an `Out` arrow as denoting a "ket" index and an `In` arrow as a "bra" index. The arrows of QN indices play two important roles in working with QN ITensors:

- A pair of QN indices must have *opposite* arrow directions to be contracted.

- When computing the QN flux of an ITensor block, QNs corresponding to an `Out` Index are added and QNs corresponding to an `In` Index are subtracted.

Examples of these arrow and flux rules will be given in the next section on QN ITensors.

## 8.3 QN ITensor

Constructing an ITensor from QN Indices makes it a QN ITensor, with a block sparse storage type. In addition to the block sparse real and complex storage types, there are also diagonal block sparse storage types which are usually obtained from factorizations such as the SVD of block sparse ITensors.

In most respects, working with QN ITensors is quite similar to working with dense ITensors. Operations like adding QN ITensors or multiplying them by scalars work in a straightforward way. However, one small but important difference from dense ITensors arises when contracting QN ITensors: matching QN indices must have opposite arrow directions to be contracted. This rule is important for consistent bookkeeping of QN flux under Hermitian conjugation of ITensors. But because computing the Hermitian conjugate `dag(T)` of a QN ITensor `T` is defined to reverse all of the arrows of its indices, code which is already written correctly for complex, dense ITensors (with proper use of `dag` to handle complex conjugation) will automatically be correct in terms of QN conservation too.

Having discussed all of the types involved in the QN ITensor system, let us discuss some examples which integrate all of these elements. An example motivated by physics is the Hilbert space of a single "hard-core" boson: a type of particle which cannot share an orbital or site with another boson. Such bosons can be used to model atoms which have large, short-range repulsive interactions. The Hilbert space of a single hard-core boson is spanned by two basis states $|0\rangle$ and $|1\rangle$, representing no particle and one particle. Along with these basis states, one can define the elementary operators $a$, $a^\dagger$, $n$, which lower, raise, or measure the number of particles:

$$
\begin{aligned}
a|1\rangle &= |0\rangle \\
a^\dagger|0\rangle &= |1\rangle \\
n|0\rangle &= 0 \\
n|1\rangle &= |1\rangle
\end{aligned}
\tag{3}
$$

Diagrammatically the equation $a|1\rangle = |0\rangle$ can be expressed as

where in the diagram note that the tensors now have arrows on their indices, with contracted indices having opposite arrow directions (`In` versus `Out`). Within ITensor, we can represent the Hilbert space of this boson as an `Index`

```
s = Index(QN("N",0)=>1,
          QN("N",1)=>1;
          tags="Boson")
```

This `Index` is the representation in code of the index lines ↗ in the $a|1\rangle = |0\rangle$ diagram above. By default, Index objects have an `Out` arrow direction meaning a contravariant index.

We can next construct the operator $a$ as the following ITensor

```
a = ITensor(s',dag(s))
a[s'=>1,s=>2] = 1.0
```

The first line constructs `a` as an ITensor with indices `s'` and `dag(s)` with elements all zero, and the second line sets the only non-zero element of `a`. We can visualize the resulting tensor as follows

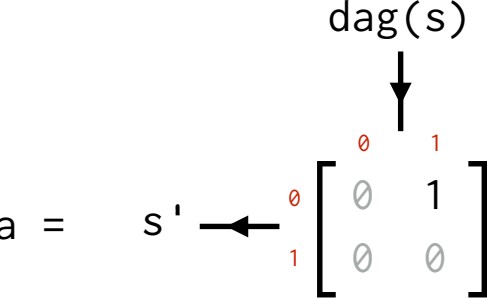

The small, red labels above denote the subspaces of the Index `s` by labelling them according to the value of the `"N"` quantum number. We can also see the single non-zero element corresponding to the (1,2) entry of the tensor and having the value 1.0.

A key point about the example of the ITensor for the $a$ operator is that *the only element stored in memory is the one shown above*. All other entries shown in light gray are assumed zero and not stored in memory. To see why this is the case, let us label each block of the $a$ tensor (or any tensor having the same indices as $a$) by its quantum number flux:

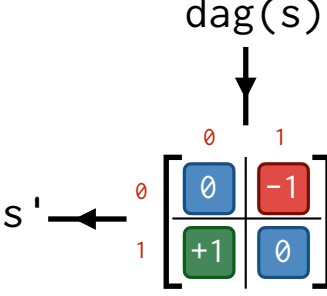

The non-zero element of the tensor $a$ is in the block with flux `QN("N",-1)` and physically means this operator always reduces the particle number by 1. Because the convention in ITensor is that QN-conserving ITensors must have a well-defined flux, only blocks with the same flux are stored in memory and the rest are assumed to be zero and not stored. In contrast, the $n$ operator has a flux of zero, and therefore will have two allowed blocks: the blocks labeled 0 and shown in blue in the diagram above.

Unlike the examples above, general QN-conserving ITensors will have many blocks which can be non-zero, and having block sizes greater than $1 \times 1$. Summations, factorizations, and especially contractions of general block sparse tensors can be much faster than for dense tensors with the same index dimensions, not only because the zero (unallocated) blocks can be skipped over, but also because operations on non-zero blocks can be performed in parallel. Both the Julia and C++ implementations of ITensor use multi-core parallelism within their block sparse tensor contraction algorithm, with speedups of up to $5 \times$ observed in practical physics applications, though the speedups vary depending on the application.

Finally, all other operations available for dense tensors work for QN-conserving ITensors too, with exactly the same interface. This includes the use of combiner ITensors, factorizations such as the SVD and QR, and higher-level algorithms involving matrix product states and operators. For further reading on how various tensor operations can be implemented while respecting Abelian group symmetries and related quantum numbers, see Ref. [29].

# 9 NDTensors Library

Early on in the design of the ITensor library, a conscious decision was made to separate the high-level ITensor interface, involving "intelligent" Index objects and related features, from the lower-level parts of the code focusing on efficient contraction routines and sparse tensor storage layouts. With the port of the ITensor library to Julia, we have taken this design one step further by making the lower-level part of the library a separate submodule[6] known as NDTensors ($N$-Dimensional Tensors) which can be used and developed separately from ITensors.jl[7].

Some of the goals of developing NDTensors as a separate module include:

- Separating low-level NDTensors algorithms from high-level ITensor logic simplifies and modularizes the code and prevents bugs.

- Encouraging more community contributions to the ITensor project, since some community members may find the NDTensors interface and features more familiar and appealing, and may not prefer to work with the ITensor layer when making contributions.

- Other software besides ITensor could eventually use NDTensors as a backend, which would promote community efforts to improve tensor software and share resources. Fully realizing this possibility would require releasing it as a separate library in the future, which we plan to do.

## 9.1 Basic Interface

The NDTensors library is a full-featured, standalone library emphasizing generic, high-performance algorithms and support for a variety of sparse tensor types. Unlike the ITensor library, NDTensors has a more traditional interface where users must keep track of the ordering of tensor indices. For example, one can construct a dense tensor with dimensions $3, 7, 4$ as

---

[6]By a module and a submodule we mean a separate namespace for defining types and methods.

[7]At the time of writing this paper, the NDTensors library can only be installed by installing the ITensors library for convenience of developing the libraries in tandem, however we plan to split it off so it can be installed seperately from ITensors in the near future.

```
using ITensors.NDTensors

T = Tensor(3,7,4)
```

which by default is filled with all zeros and then set its elements as

```
T[1,2,1] = 1.23
T[3,2,3] = -0.456
```

Tensor objects are 1-indexed, similar to Julia arrays. A tensor with complex entries can be constructed as

```
T = Tensor(ComplexF64,5,4,3)
```

Contracting two Tensors is done by specifying temporary labels for tensor indices; matching labels indicate two indices are contracted while unique labels denote uncontracted indices. In the following example:

```
A = randomTensor(3,7,2)
B = randomTensor(4,2,3)
C = contract(A,(-1,1,-2),B,(2,-2,-1))
```

the label `-1` of the first index of `A` matches the `-1` label of the third index of `B`, so those two indices are contracted with each other. Likewise the third index of `A` and second of `B` share the label `-2` and are contracted. The use of negative integers to label contracted indices is not required, but is just a convention to make the code more readable.

## 9.2  Block Sparse Tensors

NDTensors provides sparse tensor types as well. An important example is block sparsity. One way to construct a block sparse tensor is as follows:

```
blockdims = ([2,2],[2,3])
nzblocks = [(1,2),(2,1)]
A = randomBlockSparseTensor(nzblocks,blockdims)
```

The code above specifies that the tensor `A` has two indices of dimension 4 (= 2+2) and 5 (= 2+3) respectively, with the first index having two subspaces of dimensions 2 and 2 and the second index having two subspaces of dimensions 2 and 3. Thus `A` has four blocks overall, because its two indices each have two subspaces. The array `nzblocks` lists which blocks of `A` can be non-zero and will be actually allocated in memory, with each tuple giving a subspace number for each index. We can visualize a typical result for the tensor `A` as follows:

$$
A \;=\; \rule[0.5ex]{1.5em}{0.5pt}\;
\begin{array}{c c}
 & \begin{array}{ccccc} 1 & 2 & 3 & 4 & 5 \end{array} \\
\begin{array}{c} 1 \\ 2 \\ 3 \\ 4 \end{array} &
\left[
\begin{array}{ccccc}
0 & 0 & -0.1 & 0.8 & 2.7 \\
0 & 0 & 1.2 & 0.9 & -0.7 \\
-0.2 & -1.0 & 0 & 0 & 0 \\
0.3 & 0.7 & 0 & 0 & 0
\end{array}
\right]
\end{array}
$$

where the zeros shown in light gray are only assumed and not actually allocated in memory.

## 9.3 Generic Index Types

A crucial feature of NDTensor is that tensors are allowed to represent their indices not just as a collection of integers or block dimensions (specifying each the dimension of each index), but as any object providing a certain index interface. This generic design allows seamless interoperation between the NDTensors library and the ITensor library, as well as making it easy to provide features such as tensor slicing.

For dense and diag storage, essentially all that is required of the container `inds` representing the indices of a `Tensor` is that one can call the function `dim` on its `nth` element. Examples of valid `inds` objects are collections of integers, collections of ITensor `Index` objects (provided an overload of the method `dim` is provided), `Dims` objects provided by the Julia Base library for indexing built-in Julia tensors, and `BlockDims` objects defined by NDTensors for indexing block sparse tensors. By default, the strides are determined by the dimensions of the indices, but can be overloaded if needed such as for tensor slicing applications. Unless an explicit set of indices is provided, `Tensor` objects default to using the `Dims` type (a tuple of integers) to represent its indices and `BlockSparseTensor` objects default to using `BlockDims`. For block sparse storage types, an overload of the `blockdim` function is required for any block index, which is used to query the size of a specified block in a specified dimension.

## 9.4 Tensor Contraction Backend

Tensor contractions are often the computational bottleneck of tensor network algorithms. Thus implementing it as efficiently as possible is critical for performance.

For contracting two dense tensors, NDTensors currently uses a strategy of permuting and reshaping the tensors into matrices, so that the contraction maps to a matrix multiplication[8]. The motivation behind this strategy is that BLAS libraries such as Intel MKL offer such high performance that the extra overhead of permuting the tensors is worthwhile. It is also important to note that the tensor permutation has a sub-leading scaling relative to the matrix multiplication, so that in the limit of large tensors the computation is dominated by the BLAS `dgemm` or `zgemm` routines. Though this strategy is a common one for tensor libraries, its implementation in NDTensors is done carefully to ensure that every case where permutation can be avoided is taken advantage of. Also if two equivalent strategies exist to permute the contracted tensors to matrices where one of the permutations is trivial, the code chooses to permute the smaller of the two tensors.

---

[8]This is sometimes referred to as the Transpose-Transpose-GEMM-Transpose (TTGT) [32] approach

The case of block sparse tensor contraction reduces to doing a set of smaller, dense tensor contractions on various pairs of blocks from the tensors being contracted [9]. Thus it is built on top of the dense contraction layer of NDTensors, but also offers an excellent opportunity to exploit parallelism, since contraction of the blocks can be done independently, although one does have to handle cases where multiple block pairs contribute to the same block of the resulting tensor. By exploiting multi-core parallelism for the same algorithm within the C++ implementation of ITensor we have observed speedups of $2-3\times$ for DMRG and related MPS calculations (depending on the sparsity and block sizes involved, which varies strongly based on the symmetries used), and up to $5\times$ for tree tensor network calculations. More recently, we have implemented the same kind of multi-core parallelism in the block sparse contraction algorithm in NDTensors using Julia's native multithreading and have seen similar speedups to those we see in the C++ implementation that uses OpenMP.

Looking ahead, a key improvement to NDTensors will be to offer support for more advanced tensor contraction algorithms that have been recently developed. These algorithms build on sophisticated research into BLAS software, where it was realized that modern BLAS implementations could apply to the case of tensors of arbitrary order, and not just matrices. The two implementations of this type we are aware of are TBLIS [34] and TCL/GETT [32]. These libraries significantly reduce, if not totally eliminate, the permutation overhead inherent to the permute-to-matrix strategy discussed above, offering superior performance to the default contraction algorithm of C++ ITensor and NDTensors [35]. We currently have an experimental feature in the Julia version of ITensors.jl that provides TBLIS as an optional contraction backend, and have seen speedups over our current contraction code, particularly when contracting larger tensors. We plan to do benchmarks using this TBLIS backend for more sophisticated algorithms like DMRG in the near future.

# 10 Other Features of ITensor

The ITensor library has many other features which are important for productive programming, developing new algorithms or treating new problem domains, but whose precise details are somewhat beyond this high-level introduction. In this section we briefly highlight these features.

## 10.1 Writing and Reading ITensor Objects with the HDF5 Format

One important feature is that nearly every type involved in a tensor network, from Index objects to IndexSet's to ITensors, MPS, and MPOs can be written to and read from HDF5 files. The HDF5 format is a widely used and standardized format for writing large datasets and heterogenous data. It offers portability across operating systems with different binary formats; metadata and a file-system structure for organizing and retrieving data; and efficient use of memory including compression of numerical data. ITensor objects written

---

[9]Currently the default in ITensor is that block sparse tensors are contracted directly without first reshaping into a matrix. An alternative is to first permute and reshape the block sparse tensors into block sparse matrices. With that strategy, degenerate quantum number blocks can be combined, leading to a contraction involving a smaller number of larger blocks, which is advantageous for BLAS [33]. This alternative contraction strategy can be enabled with the experimental `ITensors.enable_combine_contract()` function which enables a global flag. Currently we find that neither of the two strategies (contracting versus combining then contracting) is better in every situations, and it depends on details like the quantum numbers, sparsity and order of the tensors being contracted.

```julia
using ITensors
import ITensors: op  #allows overloading of ITensors.op

op(::OpName"Sz",::SiteType"S=3/2") = [
  +3/2   0     0     0
     0   +1/2   0     0
     0    0   -1/2   0
     0    0     0   -3/2
]

op(::OpName"S+",::SiteType"S=3/2") = [
     0   sqrt(3)  0     0
     0     0      2     0
     0     0      0   sqrt(3)
     0     0      0     0
]

op(::OpName"S-",::SiteType"S=3/2") = [
     0     0     0     0
 sqrt(3)   0     0     0
     0     2     0     0
     0     0   sqrt(3)  0
]
```

Listing 1: Overloads of the `ITensors.op` method which define custom mappings of operator names to ITensors for Index objects having the tag "S=3/2".

to HDF5 files can be both written to and read from both the C++ and Julia versions of ITensor, allowing users with large C++ codes to use the new Julia version for tasks such as performing analysis of simulation results.

## 10.2  Defining Custom Local Hilbert Spaces

An important feature for physics applications is the ability to define custom "degrees of freedom" or local Hilbert spaces and associated local operators to allow users to implement their own systems of interest within high-level tools like OpSum. ITensor includes built-in definitions for only a handful of common cases such as $S = 1/2$ and $S = 1$ spin degrees of freedom, spinless and spinful fermions, and the Hilbert space of the $t-J$ model. But physics applications of ITensor often call for other definitions, such as of local Hilbert spaces for bosons, higher spin moments such as $S = 3/2$, and more exotic degrees of freedom such as $\mathbb{Z}_N$ parafermions. Users may also want to extend built-in Hilbert space types by defining additional local operators. The C++ version of ITensor already lets users define custom local Hilbert spaces and operators, but due to limitations of the C++ language the customization process has remained cumbersome and users have often had trouble mastering the necessary tasks of defining C++ types, constructors, and methods.

Fortunately, in the Julia version of ITensor we have been able to streamline the process of defining and using custom Hilbert spaces. The key innovation is that certain Index tags can be designated as special by defining associated "site types". For example, say a user

wants any Index carrying the tag `"S=3/2"` to be interpreted as a $S = 3/2$ spin (the Index should also have the appropriate dimension of 4). Practically this means we want systems such as OpSum to know how to make the appropriate local operators such as `"Sz"`, `"S+"`, and `"S-"` which act on the Hilbert space of this Index. To tell ITensor how these operators should be defined, a user can create overloads of the `ITensors.op!` method which accept a special type: `SiteType"S=3/2"`. Examples of such overloads are shown in Listing 1 and can be defined outside the ITensor library in user code. The notation `SiteType"S=3/2"` is a convenient Julia macro syntax which is used to create a unique type parameterized by a string. Creating types out of values allows one to effectively overload functions over different values, even though technically functions can only be overloaded over different types.

After defining these functions, the following code will return `"Sz"`, `"S+"`, and `"S-"` operators as ITensors given an Index s which has the `"S=3/2"` tag

```
s = Index(4,"S=3/2") # make an Index with the tag "S=3/2"
Sz = op("Sz",s)
Sp = op("S+",s)
Sm = op("S-",s)
```

The ITensor library reads the tags of the Index passed as the second argument to `op`, then checks if any of these tags have an associated `SiteType` overload of `ITensors.op`. If exactly one tag and operator name pair does have an `ITensors.op` method defined for it, such as the `::SiteType"S=3/2"`, `::OpName"Sz"` overload in Listing 1 above, then that overload is called to produce the operator corresponding to the requested name as an ITensor. Users can also overload other functions which both construct and return the operator ITensor, giving more control over the whole process.

What makes this system powerful is that the same `op` method and its overloads are called by the OpSum system and various MPS and MPO constructors within ITensor library code. So after defining the `SiteType"S=3/2"` overloads of the `op!` or `op` functions above, the following code "just works" and correctly makes an MPO of the Heisenberg Hamiltonian for an $N$-site system of $S = 3/2$ spins:

```
sites = [Index(4,"S=3/2,n=$n") for n=1:N]

os = OpSum()
for j=1:N-1
  os +=     "Sz",j,"Sz",j+1
  os += 1/2,"S+",j,"S-",j+1
  os += 1/2,"S-",j,"S+",j+1
end

H = MPO(os,sites)
```

Various special tags with associated `SiteType` operator definitions can even be mixed together in Index arrays like the `sites` array above, permitting easy setup of calculations for mixed systems such as spin chains of alternating $S = 1/2$ and $S = 1$ sites or models of alternating spin and boson sites.

## 10.3 DMRG Observer System

The DMRG code within ITensor is the most heavily used high-level feature of the library due to the continued popularity and staying power of the DMRG algorithm. Although ITensor's implementation of DMRG prints some useful details about the results of each sweep, such as the estimated energy (dominant eigenvalue) and typical bond dimension of the MPS being optimized, there are many situations where a user would like to customize the code further, such as to measure local observables throughout each sweep.

To make this customization process as easy as possible, the ITensor DMRG code accepts an optional `observer` keyword argument which allows users to pass any object which is a sub-type of `AbstractObserver`. This type should also have an overload of at least one of the methods `measure!` and `checkdone!` defined for it too. These methods can be defined in any way the user sees fit and have minimal requirements. Both are called by the ITensor DMRG code at each step of the DMRG algorithm.

The `measure!` method gets passed a variety of properties describing the current state of the DMRG calculation, such as the number of the current sweep and location of the site(s) of the MPS whose local tensors are currently being optimized, and even the entire MPS itself. A customized `measure!` function can use this information to produce a detailed snapshot of how the optimization is proceeding. One such use of the observer system in the past was to make animated movies of a DMRG calculation to be used in lectures.

The `checkdone!` method can be defined if the user wants to set some criterion for the DMRG calculation to stop before all of the requested sweeps have been performed. Example criteria could include some measure of convergence, such as the energy variance, or an external signal from the user.

# 11 Applications of ITensor

ITensor has been cited in approximately 450 research articles from 2009 to 2021.[10] Below we highlight papers which show the diverse applications of ITensor. We expect to see ever wider applications in the future as tensor network algorithms become more powerful for two- and three-dimensional systems, ab-initio Hamiltonians, and long-time dynamics [36,37], and as more applications of tensor networks are developed in applied mathematics, computer science, and machine learning [38–40].

## 11.1 Equilibrium Quantum Systems

The most common application area of tensor networks and the ITensor software to date has been equilibrium quantum systems. A common starting point for understanding equilibrium systems is through their ground state, and the DMRG algorithm which launched the field of tensor networks is primarily a ground state finding method. More recently, tensor network methods have been extended to study finite-temperature systems. Another important area of development in the field has been extending DMRG and MPS methods to handle ab initio systems such as in quantum chemistry, where details of continuum, atomic physics must be treated.

An excellent example of a ground-state study using ITensor is that of Keselman and Berg [41], who used ITensor's DMRG algorithm to compute properties of a *one-dimensional model of superconducting electrons*. A detailed study of properties of finite-size systems, including of quantities at the edge of open systems, supports the existence of a topological

---

[10]List of papers citing ITensor: https://itensor.org/papers

state of matter even in the absence of a gap in the excitation spectrum.

The state-of-the-art efficiency of ITensor's DMRG codes makes it a powerful tool for studying two-dimensional systems as well. DMRG remains one of the leading methods for studying two-dimensional quantum systems even though it scales exponentially in the transverse system size. In Refs. [42,43], Kallin, Gustainis, Johal, Stoudenmire, Melko, et al. used a combination of exact diagonalization, numerical linked cluster methods, and ITensor DMRG to obtain *entanglement entropies associated with sharp corners* in the subsystem geometry for various quantum systems at their critical points. Based on the numerical results, a conjecture was put forward for a universal scaling of this corner entanglement which was afterward supported by field theoretic methods [44].

An exemplary study using ITensor DMRG for a two-dimensional system of strongly-correlated electrons is the work by Venderley and Kim [45], who studied the hole-doped *Hubbard model on the triangular lattice*, finding a transition from p-wave to d-wave super-conductivity as the strength of on-site interactions increase.

ITensor has also been used for studying continuum electronic systems such as *quantum chemistry* calculations of hydrogen chains [20, 46, 47]; for *finite-temperature* studies, primarily in the context of the minimally entangled typical thermal state (METTS) algorithm [48–51]; and for calculations involving *PEPS two-dimensional* tensor networks [52, 53].

## 11.2 Dynamics of Quantum Systems

Dynamical behavior of quantum systems or quantum systems out-of-equilibrium is currently an active research area, where the flexibility and customizability offered by ITensor has been an excellent fit. Such customizability is important because there are many algorithms available for time-evolving quantum states [54], most of which are not totally black-box and require some care to use well. Frontier research problems also involve a variety of settings, such as closed versus open systems, or evolution via Hamiltonians versus circuits, as well as a wide range of measurements to be made of the state.

One paper typifying the use of ITensor for dynamics research, blending numerical results with theoretical predictions, is that of Alba and Calabrese Ref. [55], who showed that for *integrable systems*, such as the XXZ spin chain, one can accurately predict the entanglement entropy at both short and long times.

Nahum, Ruhman, Vijay, and Haah used ITensor in Ref. [56] to simulate dynamics of quantum states evolved by *random unitary circuits*, supporting their prediction that the growth of entanglement entropy is governed by the KPZ universality class related to the classical statistical physics of surface growth.

Schreiber et al. used ITensor to simulate the *dynamics of cold atom experiments* in Ref. [57], obtaining good agreement with experimental observations of the difference between the number of atoms in even versus odd minima of the external potential.

A rather different application of dynamical tensor network methods are as "solver" subroutines for the dynamical mean field theory (DMFT) algorithm, which can treat infinite-size systems in two and three dimensions. A novel *DMFT solver based on fork tensor network states* was proposed and demonstrated using ITensor by Bauernfeind, Zingl, et al. in Ref. [58], allowing DMFT methods to achieve greater resolution for electron spectral functions and other benefits.

## 11.3 Other Application Areas

Historically tensor network methods have mainly been developed and applied within condensed matter physics. But the recent decade has seen a major broadening in applications of tensor networks inside and outside of physics. These newer applications range

from studying holographic dualities between physical theories [59,60] to computing high-dimensional integrals in applied mathematics [38,61].

An area where tensor network methods are becoming increasingly important is *quantum computing*, where they can be used to perform efficient classical simulations of quantum devices. Tensor networks offer important advantages such as linear scaling with the number of qubits. The library PastaQ (available at github.com/GTorlai/PastaQ.jl) uses ITensor as a backend to offer tensor network methods not only for quantum simulation, but also optimization of quantum circuits, tomography of quantum systems and quantum processes, and more.

A rather different application area of tensor networks is applied mathematics and machine learning. Here tensor decomposition methods have found many different uses, from compressing weight layers of neural networks [39], to recovering missing or corrupted data using partial information [62]. Machine learning is an area where ITensor has potential to be used much more in the future, and ITensor has already been used to *investigate new models and algorithms* for machine learning, including supervised [63,64] and unsupervised [65] learning using models parameterized by tensor networks, and to investigate generalization of these models by studying synthetic data [66].

## 12 Benchmarks of ITensor Performance

To ensure that ITensor offers state-of-the-art performance, we next present benchmark results of ITensor implementations of typical tensor network algorithms and operations. One goal is comparing the performance of the C++ versus Julia implementations of ITensor, as Julia is a relatively new language whose potential for high performance computing has not yet been fully verified in every domain. Other goals of the benchmarks include testing the scaling of algorithm implementations of ITensor and showing the relative benefits of multithreading. Finally, we discuss benchmarks of ITensors versus other tensor network libraries, which we make available as an online resource, since all of the libraries involved are frequently updated and continually optimized.

All benchmarks shown here were carried out on a single workstation with four Intel Xeon Gold 6128 (Skylake) 3.4 GHz CPUs with six cores each. Times shown are "wall" or actual time, not CPU time. The BLAS and LAPACK distribution used for both the C++ and Julia calculations was Intel MKL. For the Julia ITensor benchmarks we used version 0.2.0 of ITensors.jl running on Julia version 1.6.1. The benchmarks presented below are publicly available at: https://github.com/ITensor/ITensorBenchmarks.jl.

Before we present the benchmarks, here are the high-level conclusions we draw from them:

- At least for the domain of tensor network algorithms, Julia is very competitive with C++ as a high-performance programming language.

- Some ITensor algorithms, especially those involving block sparse tensors, are currently fastest in the Julia implementation due to recent optimization efforts made there. Though most of these optimizations can be carried out in C++ too, the productivity of the Julia language and its superior libraries and tooling makes optimizations easier to identify and implement.

We again emphasize that the Julia version of ITensor is written entirely in the Julia language, without needing to perform any low-level operations in systems languages such as C++ as is often necessary in languages like Python to achieve high performance. Of course certain external libraries we use, such as BLAS and LAPACK, are written in other

languages such as Fortran, but such libraries are standard and widely used by many tensor libraries including both the C++ and Julia implementations of ITensor.

## 12.1 Comparison of Julia and C++ Implementations of ITensor

First we present a set of benchmarks comparing the performance of the C++ and Julia implementations of ITensor on seminal tensor network algorithms.

As a first comparison between the C++ and Julia implementations of ITensor, a simple but powerful tensor network algorithm is the tensor renormalization group (TRG) [67,68], which computes properties of classical statistical mechanics models at finite temperature through a decimation procedure. Each step of TRG essentially consists of contracting four tensors together into a single tensor, then performing a truncated factorization of that tensor. Below we present benchmarks of the TRG algorithm in the C++ and Julia version of ITensor, using dense tensors only, and showing calculations with 1, 4, and 8 threads used by the BLAS library within the tensor contraction steps and for different maximum bond dimensions used during the truncation steps:

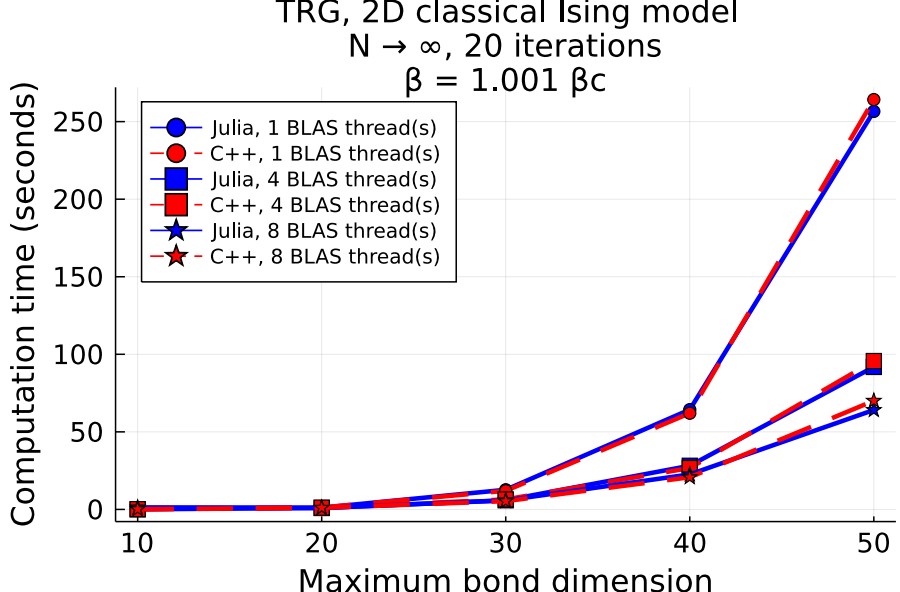

From the results, we can see that the C++ and Julia implementations have very similar performance, with the C++ version performing slightly better at bond dimension 40 and the Julia version performing better at bond dimension 50. The BLAS and LAPACK threading is clearly effective for speeding up these contraction-dominated calculations.

Another algorithm used to study classical statistical models, as well as to contract infinite PEPS tensor networks, is the corner transfer matrix renormalization group (CTMRG) [69–71]. The CTMRG algorithm decimates a contracted network of tensors by absorbing bulk tensors into boundary tensors and computing new boundary tensors at each step. Below we show the benchmark results for CTMRG using dense tensors:

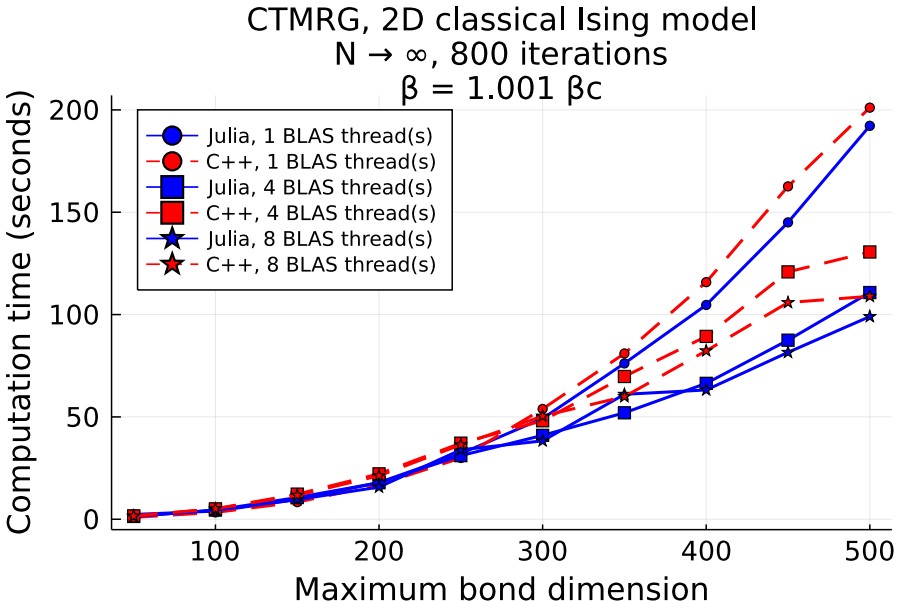

Here the Julia implementation is consistently faster for a wide range of larger bond dimensions of the boundary tensors. Allowing the BLAS to use four threads gives a speedup, but using eight threads gives little additional speedup. The relative performance as a function of BLAS threads is similar between the C++ and Julia codes, showing how the effectiveness of BLAS multithreading is dependent on the system studied and algorithm used. Speedups of the Julia versus the C++ calculations are likely due to improved dense tensor permutation libraries, specifically Strided.jl, used in the Julia version.

Now we turn to benchmarks of the density matrix renormalization group (DMRG) algorithm. DMRG calculations are the most common application of the ITensor library. We will also use DMRG as a setting to study the effect of conserving quantum numbers, resulting in block sparse tensors.

First we benchmark the simplest application of DMRG: a one-dimensional spin chain, with no quantum number conservation, that is, dense tensors:

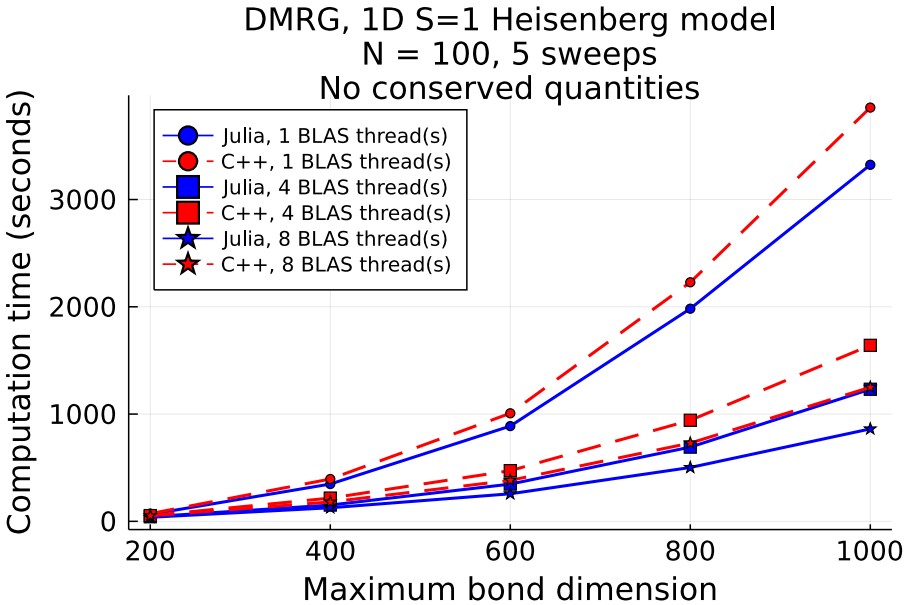

The relatively better performance of the Julia version over the C++ implementation is similar to that for CTMRG, which is sensible as the details of both algorithms are similar.

Next we consider DMRG for the same system, but conservation of the total $S^z$ spin quantum numbers and taking advantage of the resulting tensor block sparsity:

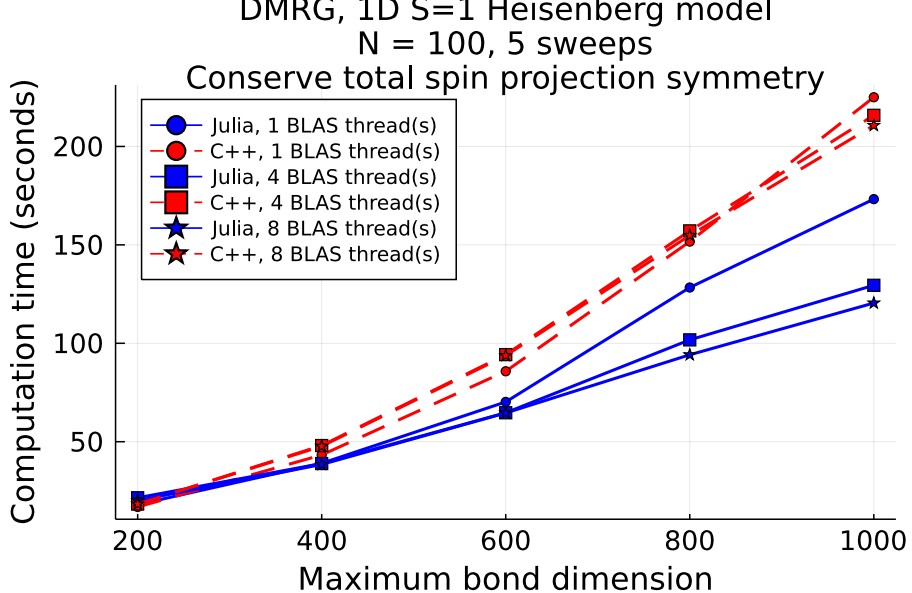

The results above show that the handling of block sparse tensors is currently much more efficient in the Julia version of ITensor versus the C++ version. This is the result of an extensive recent optimization effort, using techniques such as storing the locations of the non-zero blocks in a dictionary data structure instead of an array and optimizing contractions of small blocks. An interesting contrast of block sparse calculations versus dense calculations is that BLAS multithreading is much less effective in the block sparse case, which is likely because many of the blocks are much smaller than the overall tensor dimension, leading to smaller matrices being multiplied at the BLAS level.

Finally we benchmark the DMRG algorithm for a quasi-two-dimensional system treated by wrapping an MPS on a cylinder. Here we use the example of the Hubbard model with $U/t = 8$ and conservation of both the total $S^z$ and particle number symmetries:

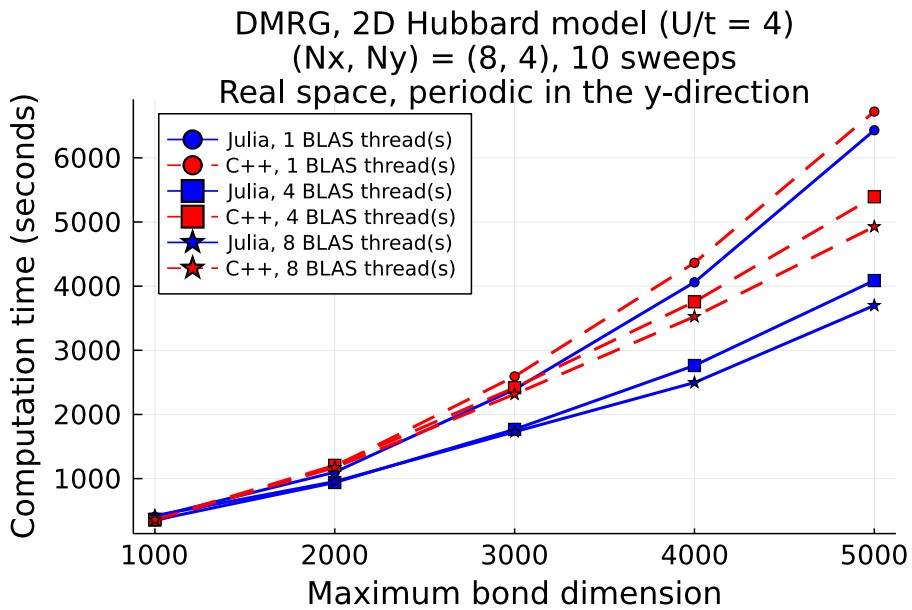

While the Julia version also outperforms the C++ version for this system, the single-threaded case is similar for both code versions, perhaps due to certain larger non-zero tensor blocks.

A technique to sparsify the tensors more in the context of two-dimensional DMRG calculations is to also conserve the momentum quantum number $k_y$ in the y-direction, or periodic direction around the cylinder [72]. By using that technique in the following benchmarks of the same two-dimensional Hubbard system, we can see that the overall time needed is reduced and the better-optimized block sparse operations in the Julia version give an even larger advantage:

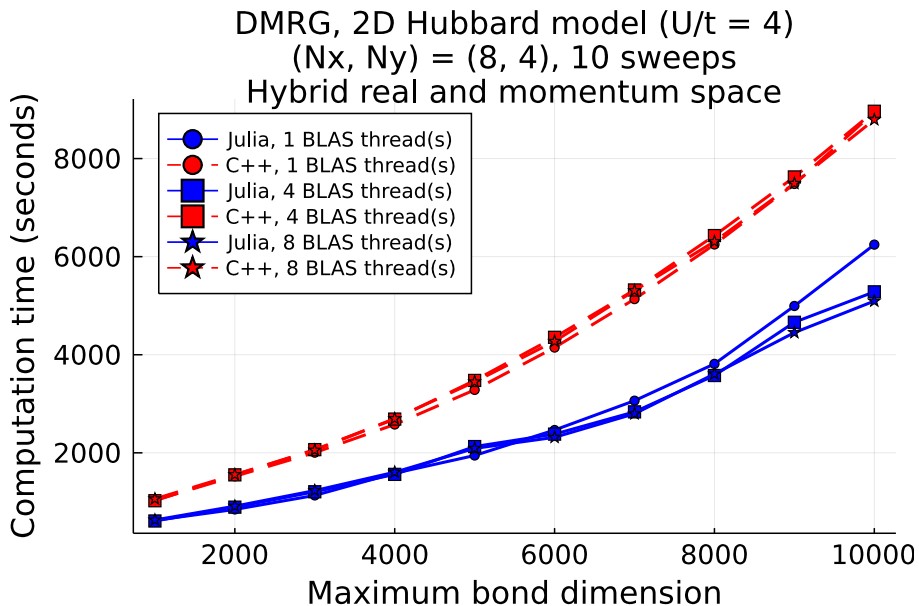

Finally, the block-sparse structure of quantum-number conserving tensors gives an opportunity for performing contractions of the non-zero blocks in parallel. We offer multithreading over block-sparse tensor contractions in both the C++ and Julia versions of ITensor. Turning on this feature and using different numbers of threads for 2D DMRG calculations gives the following timings:

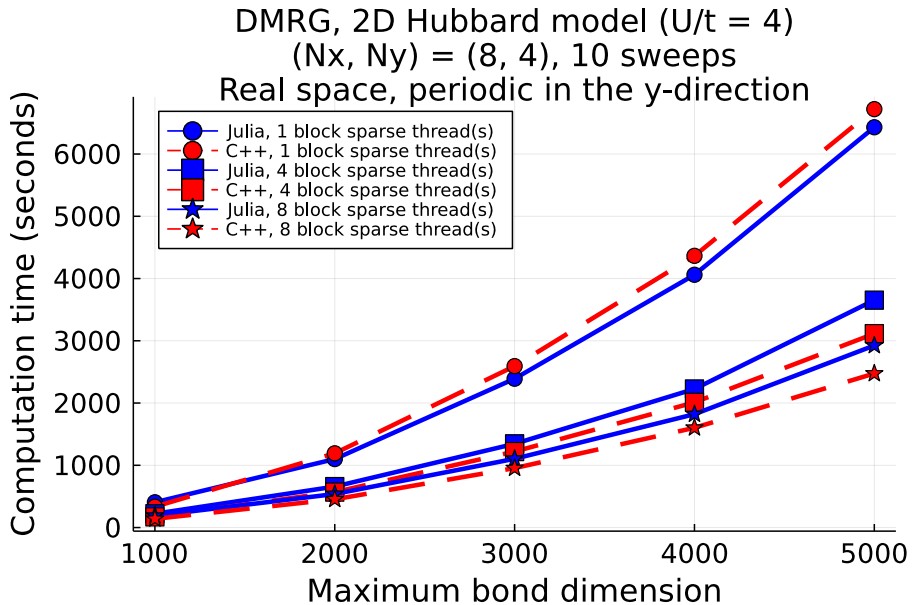

where we see a speedup of between 1.5x to 2x compared to case using no block-sparse threading. Though the single-threaded Julia implementation is slightly more efficient, the multithreading is more effective in the C++ implementation, possibly because the native Julia multithreading has a higher overhead than the OpenMP multithreading we use in C++. We plan to investigate the discrepancy in more detail.

To conclude this section, we note that the C++ implementation of ITensor, including both its tensor contraction routines and implementations of algorithm such as DMRG are already highly optimized, nearing state-of-the-art performance. So the even better performance of the Julia version of ITensor is a non-trivial outcome. The Julia version was originally modeled on the C++ implementation, but recent optimization efforts supported by Julia's more productive programming environment currently put it well ahead.

## 12.2   Benchmarks of ITensor Versus Other Software

It is important to determine how the performance of ITensor compares to other leading software. For this purpose, we have performed benchmarks comparing the Julia version of ITensor to the TeNPy high-performance tensor network library, which is implemented in a combination of Python and C++ with a Python interface [73].

However, because both ITensor and TeNPy are continually being optimized and developed, and due to subtleties of comparing different implementations of algorithms such as DMRG, we have opted not to present a definitive set of benchmarks here, but rather to host these on an external site where the results and underlying codes can be periodically updated. The latest TeNPy and ITensor benchmarks can be viewed at the following link: ITensorBenchmarks TeNPy and ITensor Comparisons.

To summarize the results of this ongoing benchmark effort, we found first of all a number of implementation differences that can inform the design and default choices of each library. For example, TeNPy by default uses a sparse representation of the Hamiltonian which we found typically speeds up DMRG significantly, so we have now implemented a similar capability in ITensor through a function called `splitblocks`, though whether using leads to better performance depends on the system, so we have not currently made it the default. Another difference is that TeNPy's DMRG implementation (as of version 0.8.4) performs more Lanczos steps within each step of DMRG compared to ITensor, which generally results in longer running times for a fixed number of DMRG sweeps. But this number is only a default setting and can be adjusted by the user. Once the algorithmic details and external dependencies (such as the BLAS library used) were made as similar as possible, we found both libraries gave comparable performance.

In the future, we plan to not only expand the set of algorithms used in the benchmark, but also to set up an automatic benchmarking system, and to include other software in the comparisons.

## 13   Future Directions

Although the ITensor library already offers high performance and powerful features for implementing any tensor network algorithm, many improvements and optimizations are planned or already under way. Here we discuss the main features under development, though some may take a different form when implemented.

A high-priority feature is support for automatic differentiation (AD). This technique has been popularized for applications in machine learning and neural networks, but has recently been demonstrated to work well for tensor networks too. For example, AD can be used for state-of-the-art infinite PEPS calculations and for calculating critical properties

of classical systems [74]. In addition, it has proven useful for optimizing tensor networks with unitary/isometric constraints like quantum circuits, MERA, and gauged MPS [75–78] as well as for computing excitations and structure factors of MPS and PEPS [79,80]. The unique index system and generic high level interface makes ITensor ideal for defining differentiation through a variety of ITensor operations. Julia's ChainRules.jl [81,82] package can be used to define basic reverse and forward mode differentiation rules independent of the particular AD framework. In conjunction with source-to-source AD frameworks available in Julia such as Zygote.jl [83] which has high coverage for differentiating through most native Julia language features, a basic ITensor AD system involving differentiating through a surprising number of ITensor operations can be written in only a few lines of code. Our use of ChainRules will allow us to target next generation AD systems being developed in Julia such as Diffractor.jl. Using this system, we have prototypes for using AD to optimize a variety of tensor network applications, such as gradient optimization of MPS, variational circuit optimization, and PEPS. We plan to extend our set of rules and coverage of ITensor operations (for example better support for differentiating tensor factorizations and MPS/MPO operations), incorporate high level support for using AD to optimize ITensor networks with unitary constraints, etc. In addition, we are investigating adding features for computing higher order derivatives of tensor networks using backends like AutoHOOT [84].

Another high-priority feature is automatic support for fermionic Hilbert spaces. Systems of fermions are foundational for physics applications of tensor networks, and are the most common type of system studied in condensed matter physics. Currently, the only automatic support for fermions in ITensor is within the OpSum/AutoMPO system, which relies on lookup tables of operator names designated as anti-commuting. That approach works well for many matrix product state calculations, but leads to a confusing experience for users when some parts of the library handles fermions automatically yet other parts of the calculation require manually introducing Jordan-Wigner string operators, such as when computing certain correlation functions or when using higher-dimensional networks such as PEPS. We are therefore experimenting with a system that introduces fermionic properties at the level of tensor indices, where index permutations result in a minus sign if odd-parity QN subspaces undergo an odd-parity permutation. Our ambitious goal is for calculations involving fermions to work with exactly the same code as for bosonic degrees of freedom. Even if some manual steps are occasionally required, this new fermion system could still be very useful.

Following the completion of the fermion system, support for other types of symmetries and non-trivial vector spaces is an important future direction. In particular, support of non-Abelian symmetries such as $SU(2)$ will be a very powerful feature for variants of the Heisenberg and Hubbard models and for electronic structure Hamiltonians such as in quantum chemistry applications.

More sophisticated optimizations of tensor contraction sequences is another future direction for ITensor. We currently have a backend for optimizing the contraction sequence of ITensors, for example to determine that the optimal sequence of a contraction like `A*B*C*D` is `(A*(B*C))*D`, based on the algorithm introduced in Ref. [85]. This can be enabled for every contraction with a global flag or for a specific contraction with a keyword argument, and additionally a custom sequence can be provided of the form `[[1,[2,3]],4]`. We are also developing tools for visualizing tensor networks which are enabled by annotating a tensor contraction with a macro, for example `@visualize A*B*C*D`. We plan to provide a variety of backends, such as a text output and an interactive output based on Makie.jl [86].

We are also developing tools for visualizing tensor networks which are enabled by annotating a tensor contraction with a macro, for example `@visualize A*B*C*D`. We

plan to provide a variety of backends, such as a text output and an interactive output based on Makie.jl [86]. This will make it easier to visualize a contraction sequence and debug code. We would like to provide alternative contraction sequence optimization backends like CoTenGra [87] which could be used to find contraction sequences for larger tensor networks than our current implementation. In addition, we are investigating incorporating general approximate contraction algorithms like those introduces in Refs. [88, 89].

We soon plan to offer first-class support for infinite MPS and MPO algorithms, with preliminary work nearly completed in the currently separate package ITensorInfiniteMPS.jl. This will include the latest developments in obtaining dominant and sub-dominant eigenvalues and MPS eigenvectors of infinite MPOs, using algorithms such VUMPS [90] and MPS tangent-space methods [27], as well as obtaining canonical forms of infinite MPS and MPOs and applying infinite MPOs to infinite MPS [91]. This will all be offered with the same level of convenience as the currently available finite MPS and MPO methods, including an infinite version of OpSum/AutoMPO.

We plan to continue developing GPU support throughout the library. Currently, only dense tensor operations can be performed on GPU, so an initial goal will be to support block sparse tensor operations on GPU. More broadly, we plan to make GPU support a first-class feature, with the eventual goal that most code written for ITensors on CPU can work directly for ITensors on GPU with high performance and minimal user effort, including code that uses automatic differentiation.

Last but not least, we hope to offer more high-level features for PEPS (two-dimensional tensor network) calculations. Algorithms and methods for optimizing PEPS have reached a point of maturity such that there are now a handful of essentially standard approaches, such as variational iPEPS [74,92,93] and fixed-point methods for computing PEPS environment tensors [71]. Many of these algorithms will be provided with ITensor in the future, and in particular leverage tools we are developing for a general tensor network interface, automatic differentiation, and contraction sequence optimization.

# Acknowledgements

We thank Johannes Hauschild for many discussions about the TeNPy software and for taking significant time to work with us to provide and develop benchmark codes. We thank Nils Wentzell for providing expertise and help regarding a custom Python environment on the Flatiron Institute computing cluster, as well as help designing the multithreading strategy for threaded block sparse contractions.

Key contributors to ITensor include: Katharine Hyatt for developing a GPU-accelerated backend for the ITensors.jl package; [11] Anna Keselman for contributing a major improvement to the OpSum/AutoMPO system which handles long-range interactions and multisite operators; Thomas E. Baker for expanding and improving the ITensor documentation, in particular the tutorials. Thanks to Jing Chen, Ying-Jer Kao, John Terilla, and Tyler Bryson for discussions about automatically handling fermion signs. We also thank Jing-Guo Liu for helping us to generalize the tensor contraction backend of ITensors.jl to handle more arbitrary number types.

Significant contributions and bug fixes to the C++ version of ITensor were made by Anna Keselman, Mingru Yang, Jack Kemp, Kyungmin Lee, Tatsuto Yamamoto, Juraj Hasik, Benedikt Bruognolo, Jose Lado, Hoi Hui, Lars-Hendrik Frahm, Lucas Vieira, Markus Wallerberger, Miles Chen, Yevgeny Bar-Lev, Jessica Alfonsi, Chuang Xi, and Andrey Antipov. We would also like to thank Nils Wentzell, Alex Wietek, and Daniel Bauern-

---

[11]ITensorGPU: https://github.com/ITensor/ITensors.jl/tree/main/ITensorGPU

feind for their help designing and testing block sparse multi-threading with OpenMP.

Significant features and bug fixes to the initial release of ITensors.jl (the Julia version of ITensor) were contributed by Katharine Hyatt, Ori Alberton, Christopher White, Jan Schneider, Alvaro Rubio-Garcia, Yiqing Zhou, Michael Abbott, Nicolau Werneck, Michael Sven Ferguson, Nick Robinson, and Amartya Bose.

**Funding information**    SRW acknowledges the support of the U.S. Department of Energy under grant DE-SC0008696. ITensor was initiated through the generous support of the DOE under award DE-SC0008696 and the NSF under award DMR-1812558, both of which continue to support the efforts of Steven R. White and his group. We are grateful for ongoing support through the Flatiron Institute, a division of the Simons Foundation.

# A  Full Code Examples

In addition to the code examples below, we include an extensive and growing set of examples as part of our source code distribution at the following link: ITensor Code Examples.

## A.1  Contraction Example

To show a fully working example of contracting two ITensors with a complicated index structure, consider the following code[12]:

```julia
using ITensors

function main()
  a = Index(3,"a")
  b = Index(2,"b")
  c = Index(4,"c")
  d = Index(5,"d")
  i = Index(2,"i")
  j = Index(6,"j")

  A = randomITensor(a,b,d,c)
  B = randomITensor(i,d,j)

  C = A * B

  @show hasinds(C,a,b,c,i,j)

  return C
end

main()
```

---

[12]Collections of indices can be made with a more compact syntax `a,b,c,d,i,j = Index.((3,2,4,5,2,6),("a","b","c","d","i","j"))`, which makes use of Julia's built in broadcast (.) syntax.

The contraction computed by this code can be expressed by the following diagram:

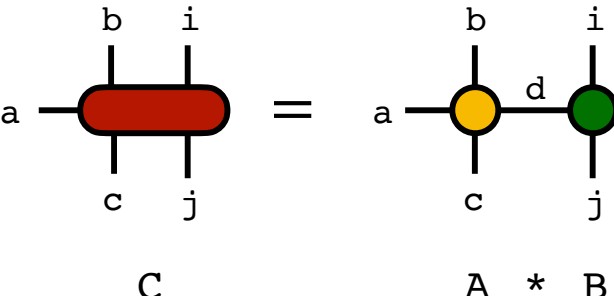

Note that the `Index` tags such as `"a","b","c"`, etc. are not required for this code to function properly, but in this context are just for making the indices easier to identify when printed.

The line of code

```
@show hasinds(C,a,b,c,i,j)
```

shows the output of the `hasinds` function which checks that the ITensor `C` has all of the indices `a,b,c,i,j`. The code above will output

```
hasinds(C,a,b,c,i,j) = true
```

## A.2   DMRG Example

The following code example shows the use of higher-level features of the ITensor Library to compute the ground-state wavefunction of the $S = 1/2$ Heisenberg quantum spin chain model using the density matrix renormalization group (DMRG) algorithm:

```
using ITensors

function main(N)
  sites = siteinds("S=1/2",N)

  os = OpSum()
  for j=1:N-1
    os += "Sz",j,"Sz",j+1
    os += 1/2,"S+",j,"S-",j+1
    os += 1/2,"S-",j,"S+",j+1
  end
  H = MPO(os,sites)

  psi0 = randomMPS(sites; linkdims=10)

  sweeps = Sweeps(5)
  setmaxdim!(sweeps, 10,20,100,100,200)
  setcutoff!(sweeps, 1E-11)

  energy, psi = dmrg(H,psi0, sweeps)
  println("G.S. energy = $energy")
  return energy, psi
end

energy, psi = main(100)
```

A typical output of this code is:

```
After sweep 1 energy=-44.062476890249 maxlinkdim=10 time=4.819
After sweep 2 energy=-44.123591549762 maxlinkdim=20 time=0.304
After sweep 3 energy=-44.127657130701 maxlinkdim=79 time=1.631
After sweep 4 energy=-44.127738543656 maxlinkdim=100 time=4.357
After sweep 5 energy=-44.127739882502 maxlinkdim=139 time=5.997
G.S. energy = -44.127739882501665
```

where note that the longer time in the first sweep includes compilation time. Brief explanations of the major steps of the above code are:

- Construct an array of $N = 100$ `Index` objects corresponding to $S = 1/2$ spins (which are dimension-2 `Index` objects labeled by the tag `"S=1/2"`).

- Input the terms of the one-dimensional Heisenberg Hamiltonian into an `OpSum` object.

- Construct an MPO `H` out of the `OpSum`.

- Construct a random MPS `psi0` of bond dimension 10.

- Create a `Sweeps` struct which indicates that five sweeps of the DMRG algorithm are to be performed, with various maximum bond dimensions allowed for each sweep and a truncation error cutoff of $10^{-11}$ throughout.

- Run the DMRG algorithm, which returns the ground-state energy and ground-state wavefunction MPS.

# B   ITensor Implementation and Interface in the C++ Language

In this appendix, we give code examples for the C++ version of ITensor to show the similarities to and differences from the Julia version.

## B.1   C++ Contraction Example

Here we show the same example of contracting two ITensors with a complicated index structure as in the previous Appendix section A.1. Consider the following code:

```cpp
#include "itensor/all.h"
#include "itensor/util/print_macro.h"
using namespace itensor;

int main()
  {
  auto a = Index(3,"a");
  auto b = Index(2,"b");
  auto c = Index(4,"c");
  auto d = Index(5,"d");
  auto i = Index(2,"i");
  auto j = Index(6,"j");

  auto A = randomITensor(a,b,c,d);
  auto B = randomITensor(i,d,j);

  auto C = A * B;

  Print(hasInds(C,a,b,c,i,j));
  }
```

By comparing to the Julia language example A.1, one can see that the C++ code above is very similar with the main differences being the use of `include` statements to import the library headers, the use of the C++ keyword `auto` on lines of code that result in the definition of a new variable, and semicolons terminating each line of procedural code. The last line uses a macro `Print` provided by ITensor, which has a similar behavior to the Julia `@show` macro and which in this case generates the output:

```cpp
Print(hasInds(C,a,b,c,i,j)) = true
```

## B.2   C++ DMRG Example

Here we show the same example of a DMRG calculation as in the previous Appendix section A.2. Consider the following code:

```cpp
#include "itensor/all.h"
using namespace itensor;

int main()
  {
  auto N = 100;

  auto sites = SpinHalf(N,{"ConserveQNs=",false});

  auto ampo = AutoMPO(sites);
  for(auto j : range1(N-1))
    {
    ampo += "Sz",j,"Sz",j+1;
    ampo += 0.5,"S+",j,"S-",j+1;
    ampo += 0.5,"S-",j,"S+",j+1;
    }
  auto H = toMPO(ampo);

  auto psi0 = randomMPS(sites,10);

  auto sweeps = Sweeps(5);
  sweeps.maxdim() = 10,20,100,100,200;
  sweeps.cutoff() = 1E-11;

  auto [energy, psi] = dmrg(H,psi0,sweeps,{"Quiet=",true});

  println("G.S. energy = ",energy);
  }
```

By comparing to the Julia language example A.2, one can see that the codes are again rather similar overall. Some key differences beyond the ones mentioned for the contraction example include that the site `Index` arrays ("site sets") in the C++ version include quantum number information by default, which we turn off in this example, and the `dmrg` routine outputs much more information by default, so we pass the named argument `{"Quiet=",true}`. These two parts of the code highlight a custom named-argument system developed for the C++ version of ITensor which could be more generally useful in other C++ codes and which we plan to release as a separate library in the future.

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
