# Peer review of "The ITensor Software Library for Tensor Network Calculations"

_SciPost Physics Codebases, doi:SciPost Phys. Codebases 4-r0.3 (2022) , SciPost Phys. Codebases 4 (2022)_

## Round 1 · Referee Report · Johannes Hauschild (Referee 1) · 2020-8-13

Strengths

  1. Easy to install.
  2. Well documented.
  3. The userguide is easy to read and provides the background and examples necessary to get started.
  4. The abstract high-level interface for contracting tensors is very intuitive and easy to use and already field-tested from the C++ version of the library.
  5. The library makes good and proper use of Julia language features, in particular multiple-dispatch.

Report

This work introduces the port of the ITensor Software library to the Julia programming language. The C++ version is already well established in the community and has been used in many works, some of which are highlighted in the presented userguide. With the rising popularity of the Julia programming language, I have no doubt that the Julia version of ITensor will also find many applications.
As illustrated by the strenght list aboce, the ITensor Software libary is of excellent quality and this work is basically ready for publication.

Personally, I would love to see some more higher-level algorithms like infinite DMRG/VUMPS/TEBD/TDVP/... for MPS and/or even PEPS codes, but (at least right now) this seems to go beyond the scope of the library. (To be fair, as developer of TeNPy I might also be a bit biased...). Given the plethora of such algorithms, it is essential to have a library like ITensor defining the basic tensor network tools, such that these individual algorithms can easily be coded up and new approaches can be tried.

Before fully recommending the publication, I have only a minor concern: benchmark tests are listed in the acceptance criteria for Scipost Physics Codebases. The folder /benchmarks of the code seems to include benchmark tests, and I even found a github action running them, but I can't find where the results are collected? Alternatively, could you provide a little bit of information how to run those benchmarks?
I would also be curious to see a speed comparison of the C++ vs. Julia version.

Requested changes

  1. Can the authors provide a bit of information how to run the benchmark or provide results?
  2. The /examples folder of the repository is somewhat hidden if the users ends up installing ITensor with the julia package manager (as recommended) and only reads the online documentation. I think just including those examples to the online documentation would benefit their use.
  3. Two minor corrections in the user guide:
  4. second-to-last paragraph of Sec. 7.2: "... orthogonal these previous ones"
  5. Second paragraph of Sec. 8.2: "where note that ..." The whole sentence is somewhat redundant, as this has already been mentioned in Sec. 2.3.

---

## Round 1 · Referee Report · Anonymous (Referee 2) · 2020-9-8

Strengths

1) Paper is well written and very useful as a first introduction with this library, as it gradually builds up complexity 2) The software discussed in this paper is easy to install and intuitive to use 3) The paper illustrated the most relevant/useful aspects of this software library using clear and instructive examples 4) Paper also discusses parts of the design philosophy and some aspects of the underlying implementation without becoming a complete implementation reference 5) The C++ version of the software is well accepted within the computation physics community for performing DMRG calculations

Weaknesses

1) The paper is sometimes a bit one-sided; e.g. while the intelligent indices certainly have many advantages, as advertised in the paper, there can also be downsides which are not at all addressed. 2) In a similar vain, there is sometimes little context (e.g. a discussion of similar or related software) or a lack of references to underlying theory (e.g. the use of quantum numbers or symmetries in tensor networks) 3) There are no benchmarks or other illustrations to substantiate claims of high efficiency, i.e. neither with respect to alternative libraries (TenPy, Uni10, ... ), nor between the Julia and C++ version of this library

Report

This manuscript provides a user introduction to the ITensor(s.jl) library, a library for using existing tensor network algorithms (mostly DMRG) or on top of which new tensor network algorithms can be developed. The main text focusses on the recently developed Julia version of the library in the examples, but the C++ version is briefly discussed in the appendix. The latter has already a long history and is well accepted within the community, this is also explicitly discussed in Section 11 of the manuscript. As indicated in the "strengths" bullet points, the paper is well written and provides a useful introduction to the library for new users, while at the same time discussing some of the underlying design philosophy, both in the user interface and the implementation.

The use of intelligent indices which are hidden from the user after the construction phase of the tensor is strongly advertised because of its benefits, namely its robustness against programming errors and the easy of specifying tensor contractions. However, there can also be downsides in comparison to alternative interfaces (the Einstein summation convention used in many libraries among which Numpy's einsum, or the more specific NCON convention that is popular among certain tensor network groups), which are not at all discussed. I think about the fact that the specific contraction that a certain block of code computes (e.g. a typical tensor network diagram in a PEPS or MERA simulation) is completely dependent on the specific input arguments and hard/impossible to assess from the contraction code itself. Furthermore, it is not clear if the intelligent index convention provides flexibility towards specifying the contraction order? The importance of contraction order and the many recent developments thereof are also completely glossed over.

Furthermore, to meet the acceptance criteria of SciPost Physics Codebases, it seems the following aspects should be addressed: 1) "...and highlight its added value as compared to existing software" 2) "Benchmarking tests must be provided." Currently, no discussion of alternative software nor any absolute or relative benchmarks are provided in the manuscript.

A final smaller comment is that, unlike most of the rest of the manuscript, section 9.3 is somewhat fuzzy and hard to parse. This section seems to go into specific technical details (names of data structures etc) of the implementation which are mostly skimmed over throughout the rest of the manuscript.

Requested changes

Address the specific acceptance criteria of SciPost Physics Codebases ( https://scipost.org/SciPostPhysCodeb/about#criteria)

1) "...and highlight its added value as compared to existing software" 2) "Benchmarking tests must be provided."

If possible

3) Provide some discussion on alternative interface or design choices and their pros and cons. 4) Improve clarity of Section 9.3

---

## Round 1 · Referee Report · Anonymous (Referee 3) · 2020-9-16

Strengths

  1. The Julia version is easier for users to get started than the C++ version.
  2. Good documentation and user guide.

Report

This paper is basically a documentation/user guide for the Juila port of ITensor library; therefore plenty of examples are given. However, the technical details and benchmarks are not clearly presented in the document. In particular, a simple benchmark against the C++ version will provide crucial information for future users' decision.

---

## Round 2 · Referee Report · Johannes Hauschild (Referee 1) · 2022-1-10

Report

I thank the authors for incorporating the suggested changes.
The software and manuscript now easily meet the acceptance criteria, and I recommend a publication.

Requested changes

Duplicated sentences "We are also developing tools for visualizing ... based on Makie.jl" in the section "Future Directions".

---

## Round 2 · Referee Report · Anonymous (Referee 3) · 2022-2-15

Weaknesses

None

Report

This updated version of manuscript includes an extensive benchmark in C++ and Julia version compared to TenPy. Also, NDTensor section is expanded and a link to example codes has been provided. With these improvements, this provides a well-written document for the Tensor package.

Requested changes

None

---

## Round 2 · Referee Report · Anonymous (Referee 2) · 2022-3-13

Report

The new version has accommodated the requested changes and was generally improved across the whole manuscript. It now meets the requirements to be accepted for publication.

One possible suggestion could be that it might have been useful to plot the benchmarks using a logarithmic scale for the runtime. This might show the scaling as a function of bond dimension more clearly, but more importantly, would probably allow for a clearer visualisation of the differences (between the different data sets) at small values of the bond dimension. In the current version, the results collapse at small runtimes due to the range of the scale.

---

## Round 2 · Author Response

We thank the referees and editors while we prepared our resubmission and hope it reflects their many helpful suggestions. Thank you to the referees for the detailed reviews and taking the time to review such a long manuscript. Below we list the major changes. Of course we have also made many small changes that are too numerous to list, such as updating code examples to use newer syntax reflecting updates we have made to ITensor, or updates about features we are currently working on.

Major changes:

  1. We have added two sections discussing benchmarks of ITensor. The first performs extensive benchmarks of ITensor itself, both the C++ and Julia implementations, across multiple algorithms and using various kinds of multi-threaded parallelism. The second set of benchmarks is more briefly discussed in the paper and is hosted externally. These compare ITensor to the state-of-the-art TeNPy software.

  2. We have revised the introduction to discuss more aspects about the unusual choice to have "intelligent" tensor indices, why it is helpful, some drawbacks it can present (but how to handle them). We have also contrasted the ITensor interface against other tensor libraries and their interfaces in some more depth, while trying to make the writing broad enough to be "future proof" given that other libraries are always changing and adding features.

  3. We discuss aspects of the various ITensor storage types in some additional depth, with more details of implementation and references to other parts of the paper where they are discussed more.

  4. On a related note, the NDTensors section has been expanded, including a more detailed example of a block-sparse tensor with a figure.

  5. Please note the link to further code examples online added to the end of Section 2 and beginning of Appendix A. We are continually updating and expanding these.

---

## Round 2 · List of Changes

Warnings issued while processing user-supplied markup:

  • Inconsistency: Markdown and reStructuredText syntaxes are mixed. Markdown will be used.
    Add "#coerce:reST" or "#coerce:plain" as the first line of your text to force reStructuredText or no markup.
    You may also contact the helpdesk if the formatting is incorrect and you are unable to edit your text.

Specific Replies to Referees

Referee 1

Thank you for the suggestion of adding benchmarks, which does make the paper much stronger. We have now added extensive benchmarks of the Julia versus C++ versions of ITensor, showing they have similar performance, with the Julia version being even faster in many cases.

Referee 2

Thank you for the detailed feedback, and especially for the suggestion of adding benchmarks.

Regarding point (1), we have added new material to the introduction to put ITensor in more context. Of course we feel the intelligent index design is a good one and are continually finding it to have many benefits (e.g. as we are now working on an automatic fermion system, automatic differentiation tools, etc.). But we certainly agree that there are risks or possible downsides of the design that have to be thought about: for example, we now specifically note how users can gain manual control over index ordering and memory layout when needed.

To address point (2), we have added references to relevant papers on conservation of quantum numbers i.e. symmetric tensors within the section on QN conserving ITensors. Thank you for this suggestion since there has been much work on this topic in the tensor literature of course.

Regarding (3), we have now added extensive benchmarks of the C++ versus Julia implementations of ITensor, and detailed benchmarks versus the TeNPy software online. We agree the paper greatly benefits from including these.

Referee 3

Thank you for the detailed review. In the new section on benchmarks, we have also provided links to our ITensorBenchmarks Github repository, which includes further instructions for running or re-running the benchmark codes with various adjustable options. We plan to continue working on this repository over time by not only adding more benchmarks but also making it easier to use with more adjustable options etc.

We have also provided a link to the examples code folder, plus made this more prominent in our online ITensor documentation. Thank you for this suggestion as these examples were definitely too hard to find previously.

Thank you for the writing corrections which we have implemented.

---

## Editorial Decision

published